# A LARGE DEVIATION THEORY ANALYSIS ON THE IMPLICIT BIAS OF SGD

## ABSTRACT

Stochastic Gradient Descent (SGD) plays a key role in training deep learning models, yet its ability to implicitly regularize and enhance generalization remains an open theoretical question. We apply Large Deviation Theory (LDT) to analyze why SGD selects models with strong generalization properties. We show that the generalization error jointly depends on the level of concentration of its empirical loss around its expected value and the *abnormality* of the random deviations stemming from the stochastic nature of the training data observation process. Our analysis reveals that SGD gradients are inherently biased toward models exhibiting more concentrated losses and less abnormal and smaller random deviations. These theoretical insights are empirically validated using deep convolutional neural networks, confirming that mini-batch training acts as a natural regularizer by preventing convergence to models with high generalization errors.

## 1 INTRODUCTION

Stochastic Gradient Descent (SGD) has become a crucial tool in modern deep learning, driving the training of models that power today's AI applications (Bottou, 2010). In addition to being an efficient optimization algorithm, SGD plays a vital role in shaping the generalization performance of models, particularly in overparameterized systems where many solutions can perfectly fit the training data (Zhang et al., 2017). Remarkably, SGD exhibits an implicit bias toward solutions that generalize well to unseen data, an intriguing phenomenon that has captured the attention of researchers.

A key reason for this implicit bias is the inherent noise from the stochastic nature of gradient updates in SGD (Neyshabur et al., 2015b; Zou et al., 2021). This noise directs the optimization process toward flat minima —solutions that are less sensitive to data perturbations— resulting in models with robust generalization (Keskar et al., 2016). Researchers have linked this behavior to a preference for simpler, lower-complexity solutions, indicating that SGD's stochasticity acts as an implicit regularizer (Neyshabur et al., 2015b; Hardt et al., 2016; Zou et al., 2021; Tian et al., 2023). Recent studies show that, even in simple models like linear regression, SGD's implicit regularization can outperform explicit methods like ridge regression, especially in overparameterized settings (Zou et al., 2021). These insights underscore the importance of algorithmic regularization in deep learning, yet there is still a pressing need for new perspectives and explanations to unravel the relationship between SGD, noise, and generalization. The exact nature of this phenomenon remains one of the most compelling open questions in theoretical machine learning (Ghorbani et al., 2019).

We present a novel theoretical analysis of SGD's implicit bias using principles from Large Deviation Theory (LDT) (Ellis, 2006; Touchette, 2009). We introduce a new decomposition of the generalization error based on the *rate function*, showing that it depends on the concentration of empirical loss around its expected value and the magnitude of random deviations from the stochastic training data. This decomposition breaks the gradient of the training loss into three terms: (i) biases the algorithm toward models with lower expected loss, (ii) favors less concentrated empirical losses, and (iii) promotes models with larger random deviations. We show that small mini-batches prevent SGD from converging to models with large generalization error.

These findings provide a new perspective on SGD and suggest ways to improve optimization. Specifically, we show that SGD does not need to follow every mini-batch's gradients to achieve low generalization error. By discarding mini-batches that contribute little information, we can achieve more efficient solutions with better generalization, paving the way for enhanced performance in SGD.

## 2 PRELIMINARIES

In this work, we build on the idea that the empirical loss of each model in a given class behaves as a random variable with a distinct mean and varying concentration around that mean. A dataset induces a realization of this random variable. Formally, let $D$ represent a training dataset of size $n > 0$, generated i.i.d. from an unknown distribution $\nu(\boldsymbol{y}, \boldsymbol{x})$. The model class is parameterized by $\boldsymbol{\theta} \in \Theta$, and for each model $\boldsymbol{\theta}$, its loss function $\ell(\boldsymbol{y}, \boldsymbol{x}, \boldsymbol{\theta})$ is assumed to be positive. The expected loss is $L(\boldsymbol{\theta}) = \mathbb{E}_\nu[\ell(\boldsymbol{y}, \boldsymbol{x}, \boldsymbol{\theta})]$, while the empirical loss on dataset $D$ is $\hat{L}(D, \boldsymbol{\theta}) = \frac{1}{n} \sum_{i=1}^{n} \ell(\boldsymbol{y}_i, \boldsymbol{x}_i, \boldsymbol{\theta})$. The empirical loss $\hat{L}(D, \boldsymbol{\theta})$ behaves as a random variable $\hat{L}_n(\boldsymbol{\theta})$, as it is derived from the randomly sampled dataset $D \sim \nu^n$. The realized value of $\hat{L}_n(\boldsymbol{\theta})$ when the dataset $D$ is observed is denoted as $\hat{L}(D, \boldsymbol{\theta})$. Each model's empirical loss $\hat{L}_n(\boldsymbol{\theta})$ has mean equal to $L(\boldsymbol{\theta})$, but the degree of concentration around this mean varies. Figure 1 (left) illustrates this with histograms for three InceptionV3 models (Szegedy et al., 2016), using datasets of size $n = 50$, produced using methods from Masegosa and Ortega (2024). The histograms show that concentration varies: the *Initial* model (using *Kaiming or He initialization* (He et al., 2015)) is highly concentrated around its mean $L(\boldsymbol{\theta}) = \ln 10$, while the $\ell_2$-regularized model also has greater concentration compared to the *Standard* model.

Empirical risk minimization seeks to find a model $\boldsymbol{\theta}$ that minimizes the realized empirical loss, $\min_{\boldsymbol{\theta}} \hat{L}(D, \boldsymbol{\theta})$. The main challenge is to choose models whose empirical loss is close to the expected loss $L(\boldsymbol{\theta})$, ensuring a small generalization error, defined as the difference between $\hat{L}(D, \boldsymbol{\theta})$ and $L(\boldsymbol{\theta})$. This work demonstrates that generalization error is influenced by two key factors: (i) the *level of concentration* of the random variable $\hat{L}_n(\boldsymbol{\theta})$; a small empirical loss $\hat{L}(D, \boldsymbol{\theta})$ could result from a model with a high expected loss $L(\boldsymbol{\theta})$ but poor concentration, which is undesirable since such models generalize poorly. It could also come from a model with a well-concentrated $\hat{L}_n(\boldsymbol{\theta})$ and a lower mean $L(\boldsymbol{\theta})$, the desired outcome. (ii) the *level of abnormality of the generalization error*, which refers to the possibility that a small empirical loss may be due to an unlikely, abnormal occurrence from the left tail of $\hat{L}_n(\boldsymbol{\theta})$, irrespective of the concentration level.

In order to mathematically formalize these two factors, we use the so-called rate function, the central function in LDT, which is denoted by $\mathcal{I}_{\boldsymbol{\theta}}(a) : \mathbb{R} \to \mathbb{R}$, and it is defined as the *Legendre transform* of the *cumulant generating function*, denoted by $J_{\boldsymbol{\theta}}(\lambda) : \mathbb{R} \to \mathbb{R}^+$. In this work, we introduced a signed version of the rate function and consider the *cumulant generating function* of the model's centered loss. These two functions are defined as

$$J_{\boldsymbol{\theta}}(\lambda) = \ln \mathbb{E}_\nu \left[ e^{\lambda(L(\boldsymbol{\theta}) - \ell(\boldsymbol{y}, \boldsymbol{x}, \boldsymbol{\theta}))} \right] \quad \text{and} \quad \mathcal{I}_{\boldsymbol{\theta}}(a) = sign(a) \cdot \sup_{\lambda \in \mathbb{R}} \lambda a - J_{\boldsymbol{\theta}}(\lambda), \quad (1)$$

where $\mathcal{I}_{\boldsymbol{\theta}}(a)$ is a *signed* rate function to make it invertible in $\mathbb{R}$. The rate $\mathcal{I}_{\boldsymbol{\theta}}(a)$ and the cummulant $J_{\boldsymbol{\theta}}(\lambda)$ are well defined, positive and strictly monotonic real-valued functions, satisfying $\mathcal{I}_{\boldsymbol{\theta}}(0) = 0$ and $J_{\boldsymbol{\theta}}(0) = 0$ (Rockafellar, 1970).

The relevance of the rate function is consequence of Chernoff's bound, which upper-bounds how likely is to observe an empirical loss $\hat{L}(D, \boldsymbol{\theta})$ that largely deviates from the expected loss $L(\boldsymbol{\theta})$.

**Theorem 1** (Chernoff (1952)). *For any fixed $\boldsymbol{\theta} \in \Theta$ and $n > 0$, it satisfies*

$$\begin{aligned} \forall a \geq 0, \quad \mathbb{P}_{D \sim \nu^n} \left( L(\boldsymbol{\theta}) - \hat{L}(D, \boldsymbol{\theta}) \geq a \right) \leq e^{-n|\mathcal{I}_{\boldsymbol{\theta}}(a)|}, \\ \forall a \leq 0, \quad \mathbb{P}_{D \sim \nu^n} \left( L(\boldsymbol{\theta}) - \hat{L}(D, \boldsymbol{\theta}) \leq a \right) \leq e^{-n|\mathcal{I}_{\boldsymbol{\theta}}(a)|}. \end{aligned} \quad (2)$$

On the other hand, Cramér's Theorem (Cramér, 1938) states that Chernoff's bound is exponentially tight for *large $n$*. Formally, this statement is written as follows,

**Theorem 2** (Cramér (1938); Ellis (2006)). *For any fixed $\boldsymbol{\theta} \in \Theta$ and any $a > 0$, it satisfies*

$$\lim_{n \to \infty} -\frac{1}{n} \ln \mathbb{P}_{D \sim \nu^n} \left( L(\boldsymbol{\theta}) - \hat{L}(D, \boldsymbol{\theta}) \geq a \right) = |\mathcal{I}_{\boldsymbol{\theta}}(a)|.$$

The same result holds for the left tail, $\mathbb{P}_{D \sim \nu^n} \left( L(\boldsymbol{\theta}) - \hat{L}(D, \boldsymbol{\theta}) \leq a \right)$ with $a \leq 0$. In LDT, the above asymptotic result is stated by saying the Chernoff's bound is *exponentially tight for large $n$*. Formally, there exists a function $o(n, a)$ such that $\lim_{n \to \infty} \frac{1}{n} o(n, a) = 0$, verifying

$$\forall a \geq 0, \quad \mathbb{P}_{D \sim \nu^n} \left( L(\boldsymbol{\theta}) - \hat{L}(D, \boldsymbol{\theta}) \geq a \right) = e^{-n|\mathcal{I}_{\boldsymbol{\theta}}(a)| + o(n, a)} \asymp e^{-n|\mathcal{I}_{\boldsymbol{\theta}}(a)|}, \quad (3)$$

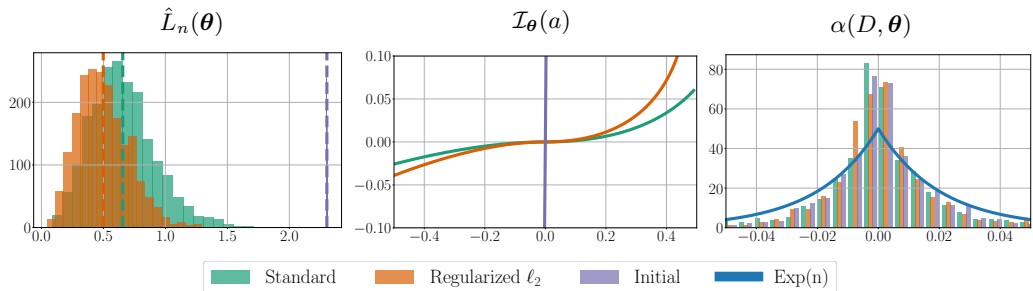

Figure 1: Visualization of the distribution $\hat{L}_n(\boldsymbol{\theta})$ (left), the rate function $\mathcal{I}_{\boldsymbol{\theta}}(a)$ (center), and the abnormality $\alpha(D, \boldsymbol{\theta})$ (right) for three InceptionV3 models trained on CIFAR-10. The models considered include a standard SGD-trained model, a $\ell_2$-regularized model, and the initial model before training. In the right panel, the Exponential of $n$ is displayed twice to illustrate Theorem 4.

where $\asymp$ denotes asymptotic equality (Ellis, 2006). The expression above demonstrates that the exact value of $\mathbb{P}_{D \sim \nu^n}(L(\boldsymbol{\theta}) - \hat{L}(D, \boldsymbol{\theta}) \geq a)$ is determined by the rate function, along with an additional sub-exponential term that becomes negligible for sufficiently large $n$. Therefore, for large $n$, the rate function effectively captures the level of concentration of the empirical loss $\hat{L}_n(\boldsymbol{\theta})$ around its expected value $L(\boldsymbol{\theta})$, because it defines the *survival function* of the random variable $L(\boldsymbol{\theta}) - \hat{L}(D, \boldsymbol{\theta})$ with $D \sim \nu^n$. As a result, models with larger rate functions are less likely to exhibit large differences between their expected and its realized empirical losses. This relationship within the context of machine learning has been recently examined by Masegosa and Ortega (2024).

Figure 1 (center) presents the rate functions for the three previously discussed InceptionV3 (Szegedy et al., 2016) neural networks, estimated using the procedures outlined in Masegosa and Ortega (2024). The rate functions clearly reflect the varying levels of concentration in the empirical losses, as depicted by the histograms in Figure 1 (left). The *Initial* model exhibits a prominent rate function, while the *Standard* model has a smaller rate function compared to the $\ell_2$-regularized model.

## 3 THE IMPLICIT BIAS OF GRADIENT DESCENT (GD)

In this section, we introduce a novel decomposition of a model's generalization error, formalize the concept of *abnormality* in the generalization error, and demonstrate how (full-batch) Gradient Descent (GD) is biased toward finding models with poorly concentrated empirical losses and whose realized empirical loss deviates abnormally from the expected loss.

### DECOMPOSING THE EMPIRICAL LOSS

The following result presents a novel decomposition of the empirical loss in terms of the expected loss $L(\boldsymbol{\theta})$, the inverse of the (signed) rate function, denoted $\mathcal{I}_{\boldsymbol{\theta}}^{-1}(s)$, and a function $\alpha : \mathcal{D} \times \boldsymbol{\Theta} \to \mathbb{R}$. As argued in the next section, $\alpha(D, \boldsymbol{\theta})$ measures the *degree of abnormality* of the observed generalization error, $L(\boldsymbol{\theta}) - \hat{L}(D, \boldsymbol{\theta})$, for the model $\boldsymbol{\theta}$.

**Proposition 3.** *For any $D \sim \nu^n$ and any $\boldsymbol{\theta} \in \boldsymbol{\Theta}$, we have that*

$$\hat{L}(D, \boldsymbol{\theta}) = L(\boldsymbol{\theta}) - \mathcal{I}_{\boldsymbol{\theta}}^{-1}(\alpha(D, \boldsymbol{\theta})) . \tag{4}$$

*where $\alpha : \mathcal{D} \times \boldsymbol{\Theta} \to \mathbb{R}$ is defined as $\alpha(D, \boldsymbol{\theta}) := \mathcal{I}_{\boldsymbol{\theta}}(L(\boldsymbol{\theta}) - \hat{L}(D, \boldsymbol{\theta}))$.*

Although the above decomposition is technically simple, it effectively breaks down the empirical loss into three distinct components with highly meaningful interpretations. The first component is the expected loss, denoted as $L(\boldsymbol{\theta})$. The second component, a composite term, measures the deviation of the observed empirical loss from its expected value, often referred to as the generalization error. Within this term, the function $\mathcal{I}_{\boldsymbol{\theta}}^{-1}(s)$ defines the level of concentration of $\hat{L}_n(\boldsymbol{\theta})$ around its expected value. As shown in Section 2, models with a high rate function exhibit greater concentration. Consequently, models with a smaller inverse rate function $\mathcal{I}_{\boldsymbol{\theta}}^{-1}(s)$ are more concentrated too. Actually, a

second-order Taylor expansion $\mathcal{I}_{\boldsymbol{\theta}}^{-1}(s)$ around $s = 0$ shows how this quantity is closely related to the standard-deviation of the loss of a model, denoted by $\sigma(\boldsymbol{\theta})$:

$$\mathcal{I}_{\boldsymbol{\theta}}^{-1}(s) \approx \text{sign}(s)\sqrt{2|s|}\sigma(\boldsymbol{\theta}), \quad \text{where} \quad \sigma(\boldsymbol{\theta}) := \sqrt{\mathbb{E}_{\nu}[(\ell(\boldsymbol{y}, \boldsymbol{x}, \boldsymbol{\theta}) - L(\boldsymbol{\theta}))^2]}. \tag{5}$$

Finally, it is important to note that both $L(\boldsymbol{\theta})$ and $\mathcal{I}_{\boldsymbol{\theta}}^{-1}(s)$ are deterministic; all the randomness in $\hat{L}(D, \boldsymbol{\theta})$ arises from the abnormality value $\alpha(D, \boldsymbol{\theta})$. As we will show in the next section, the value of $\alpha(D, \boldsymbol{\theta})$ represents the degree of abnormality in the magnitude of the generalization error. $\alpha(D, \boldsymbol{\theta})$ will be higher when the observed $\hat{L}(D, \boldsymbol{\theta})$ comes from the tails of $\hat{L}_n(\boldsymbol{\theta})$ and small otherwise. Since $\mathcal{I}_{\boldsymbol{\theta}}^{-1}(s)$ *increases monotonically with* $s$ (Rockafellar, 1970), a larger $\alpha(D, \boldsymbol{\theta})$ value leads to a greater difference between $L(\boldsymbol{\theta})$ and $\hat{L}(D, \boldsymbol{\theta})$.

THE ABNORMALITY OF THE GENERALIZATION ERROR

In this work, we propose that $\alpha(D, \boldsymbol{\theta})$, as defined in Proposition 3, serves as a measure of the degree of abnormality in an observed generalization error. A large difference between $\hat{L}(D, \boldsymbol{\theta})$ and $L(\boldsymbol{\theta})$ can be considered highly unlikely or *abnormal* if the model's empirical loss $\hat{L}_n(\boldsymbol{\theta})$ is tightly concentrated around its mean $L(\boldsymbol{\theta})$, indicating that $\hat{L}(D, \boldsymbol{\theta})$ is sampled from the tails of $\hat{L}_n(\boldsymbol{\theta})$. Conversely, the same difference between $\hat{L}(D, \boldsymbol{\theta})$ and $L(\boldsymbol{\theta})$ may *not be abnormal* for a model whose empirical loss $\hat{L}_n(\boldsymbol{\theta})$ is poorly concentrated.

Using the approximation of Equation (5), we can derive the following approximation for $\alpha(D, \boldsymbol{\theta})$:

$$\alpha(D, \boldsymbol{\theta}) \approx sign(L(\boldsymbol{\theta}) - \hat{L}(D, \boldsymbol{\theta}))\sigma(\boldsymbol{\theta})^{-2}(L(\boldsymbol{\theta}) - \hat{L}(D, \boldsymbol{\theta}))^2. \tag{6}$$

From this approximation we can start to understand why large $\alpha(D, \boldsymbol{\theta})$ values corresponds to situations where $\hat{L}(D, \boldsymbol{\theta})$ is abnormally distant from its mean $L(\boldsymbol{\theta})$. In general, according to Theorem 2, $\alpha(D, \boldsymbol{\theta})$ asymptotically equals the (normalized) log-probability of observing a generalization error higher or equal than $L(\boldsymbol{\theta}) - \hat{L}(D, \boldsymbol{\theta})$ as:

$$\alpha(D, \boldsymbol{\theta}) \asymp -\frac{1}{n} \ln \mathbb{P}_{S \sim \nu^n}\left(L(\boldsymbol{\theta}) - \hat{L}(S, \boldsymbol{\theta}) \geq L(\boldsymbol{\theta}) - \hat{L}(D, \boldsymbol{\theta})\right). \tag{7}$$

Intuitively, given a fixed dataset $D$, the abnormality rate $\alpha(D, \boldsymbol{\theta})$ *measures the log-probability of observing, for another dataset $S$, a larger generalization error than the one observed with $D$*. Using it to compare two models $\boldsymbol{\theta}$ and $\boldsymbol{\theta}'$, if $\alpha(D, \boldsymbol{\theta}) \geq \alpha(D, \boldsymbol{\theta}')$, observing, for another dataset $S \sim \nu^n$, a generalization error higher or equal than the one observed with $D$ is *more unlikely* for $\boldsymbol{\theta}$ than for $\boldsymbol{\theta}'$. We then say that *the observed generalization error for $D$ was more abnormal under $\boldsymbol{\theta}$ than under $\boldsymbol{\theta}'$*.

The following result shows how $\alpha(D, \boldsymbol{\theta})$, as a random variable over $D \sim \nu^n$, is highly related to an exponential distribution of parameter $n$:

**Theorem 4.** *For any $\boldsymbol{\theta} \in \Theta$, $n > 0$ and $D \sim \nu^n$, the cumulative of distribution of $\alpha(D, \boldsymbol{\theta})$ satisfies*

$$\forall s > 0 \quad \mathbb{P}_{D \sim \nu^n}\left(\alpha(D, \boldsymbol{\theta}) \geq s\right) \leq e^{-n|s|} \quad \text{and} \quad \forall s < 0 \quad \mathbb{P}_{D \sim \nu^n}\left(\alpha(D, \boldsymbol{\theta}) \leq s\right) \leq e^{-n|s|}, \tag{8}$$

*and both inequalities are asymptotically tight,*

$$\forall s > 0 \quad \mathbb{P}_{D \sim \nu^n}(\alpha(D, \boldsymbol{\theta}) \geq s) \asymp e^{-n|s|} \quad \text{and} \quad \forall s < 0 \quad \mathbb{P}_{D \sim \nu^n}(\alpha(D, \boldsymbol{\theta}) \leq s) \asymp e^{-n|s|}. \tag{9}$$

The result given by Equation (8) shows that the tails of the distribution of $\alpha(D, \boldsymbol{\theta})$ are always *thinner* than those of an exponential distribution with rate $n$, denoted as $Exp(n)$. Crucially, *this always happens regardless of the model or the data-generating distribution*. This insight allows us to accurately quantify the degree of abnormality in the generalization error of a model by positioning the corresponding $\alpha(D, \boldsymbol{\theta})$ value within the tail of an exponential distribution. For example, in a dataset of size 50 000, when $\alpha(D, \boldsymbol{\theta}) \geq \frac{1}{50\,000} \ln \frac{1}{0.01} \approx 0.0001$, the probability of randomly observing such an event is less than 1%. This is a *universal* cut-off, because it is applicable for any model and for any data-generating distribution.

The second result, presented in Equation (9), shows that for *large datasets*, $\alpha(D, \boldsymbol{\theta})$ closely approximates a zero-centered double-exponential distribution, or Laplace distribution, regardless of the model or the data-generating distribution. This indicates that for *large datasets, the stochasticity associated with $\hat{L}(D, \boldsymbol{\theta})$ can be effectively represented by a Laplace distribution, independently of*

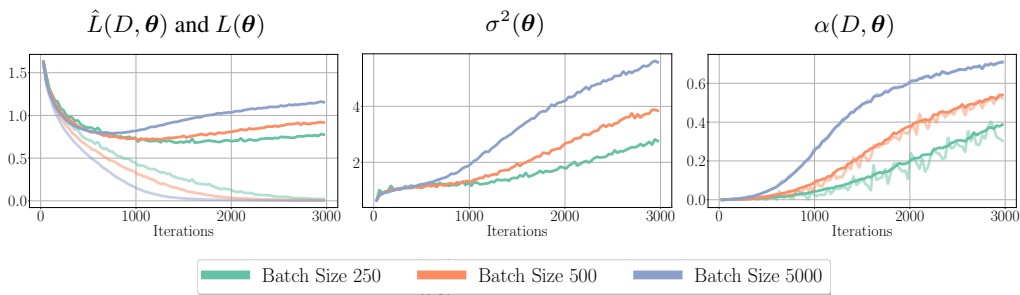

Figure 2: Evolution of training and test loss (left), loss variance (center), and abnormality rate (right) for InceptionV3 models trained with varying batch sizes. $\ell_2$ regularization is applied to the model trained with a larger batch size. In the right panel, $\alpha(B_t, \boldsymbol{\theta}_t)$ is depicted with a shadowed color to emphasize its proximity to $\alpha(D, \boldsymbol{\theta})$.

*the model family or the underlying data-generating process.* Figure 1 (right) illustrates this point with surprising accuracy. The figure shows how the empirical distribution of $\alpha(D, \boldsymbol{\theta})$ for three very different InceptionV3 models trained on Cifar10, where $D \sim \nu^{50}$, closely resembles a double-exponential or Laplace distribution, even with such as a small $n$ value. As conclusion, the distribution of $\hat{L}_n(\boldsymbol{\theta})$ for *large $n$* values can be expressed as:

$$\hat{L}(D, \boldsymbol{\theta}) \approx L(\boldsymbol{\theta}) - \mathcal{I}_{\boldsymbol{\theta}}^{-1}(s), \qquad s \sim \text{Laplace}(0, n). \tag{10}$$

The above equation resembles the reparametrization of a Gaussian distribution, particularly when considering the approximation given in Equation (5). This perspective highlights a novel asymptotic approximation of the generalization error offered by Large Deviation Theory (LDT) (Ellis, 2006), that, at first, differs from the one provided by the Central Limit Theorem.

THE IMPLICIT BIAS OF GRADIENT DESCENT (GD)

Proposition 3 sheds light on the different trade-offs involved in the minimization of the empirical loss. This result can be used to decompose the gradient of $\hat{L}(D, \boldsymbol{\theta})$ in three different terms at each iteration $t$ of the optimization process followed by GD. More precisely:

$$\nabla_{\boldsymbol{\theta}} \hat{L}(D, \boldsymbol{\theta}_t) = \nabla_{\boldsymbol{\theta}} L(\boldsymbol{\theta}_t) - \nabla_{\boldsymbol{\theta}} \mathcal{I}_{\boldsymbol{\theta}_t}^{-1}(s)_{|s=\alpha(D,\boldsymbol{\theta}_t)} - \nabla_s \mathcal{I}_{\boldsymbol{\theta}_t}^{-1}(s)_{|s=\alpha(D,\boldsymbol{\theta}_t)} \nabla_{\boldsymbol{\theta}} \alpha(D, \boldsymbol{\theta}_t). \tag{11}$$

To simplify the analysis, and without any loss of generality, we will assume through the rest of the paper that $L(\boldsymbol{\theta}_t) > \hat{L}(D, \boldsymbol{\theta}_t)$, because this is always the case in GD after very few iterations. According to Proposition 3 and the above decomposition of the gradient, when GD minimizes the empirical loss $\hat{L}(D, \boldsymbol{\theta})$ involves the minimization/maximization of the following terms:

1. $\nabla_{\boldsymbol{\theta}} L(\boldsymbol{\theta}_t)$ *points* towards models with small expected loss $L(\boldsymbol{\theta})$.
2. $-\nabla_{\boldsymbol{\theta}} \mathcal{I}_{\boldsymbol{\theta}_t}^{-1}(s)$ *points* towards models with poorly concentrated $\hat{L}_n(\boldsymbol{\theta})$.
3. $-\nabla_{\boldsymbol{\theta}} \alpha(D, \boldsymbol{\theta}_t)$ *points* towards models with abnormal generalization errors.

The third term in the decomposition is multiplied by $\nabla_s \mathcal{I}_{\boldsymbol{\theta}_t}^{-1}(s)_{|s=\alpha(D,\boldsymbol{\theta}_t)}$, which is a scalar. Since $L(\boldsymbol{\theta}_t) > \hat{L}(D, \boldsymbol{\theta}_t)$, this term is always positive and does not influence the gradient's direction.

These dynamics are clearly depicted in Figure 2, which illustrates the behavior of the gradient descent optimizer with very large batches (batch size 5 000). In Figure 2 (left), we observe that $\hat{L}(D, \boldsymbol{\theta})$ decreases monotonically, while $L(\boldsymbol{\theta})$ decreases during the first half of the iterations but then begins to slightly increase in the latter half. Figure 2 (center) displays the evolution of the variance of the model's loss function, which is a proxy to measure the degree of concentration of $\hat{L}_n(\boldsymbol{\theta}_t)$, showing a consistent increase over time. Finally, Figure 2 (right) demonstrates the progression of the abnormality rate $\alpha(D, \boldsymbol{\theta})$, which steadily rises during the entire optimization process.

It is noteworthy to see how GD converges to models whose realized empirical loss $\hat{L}(D, \boldsymbol{\theta})$ is *very abnormally* far from the expected loss $L(\boldsymbol{\theta})$. To get a sense of how much abnormal are these

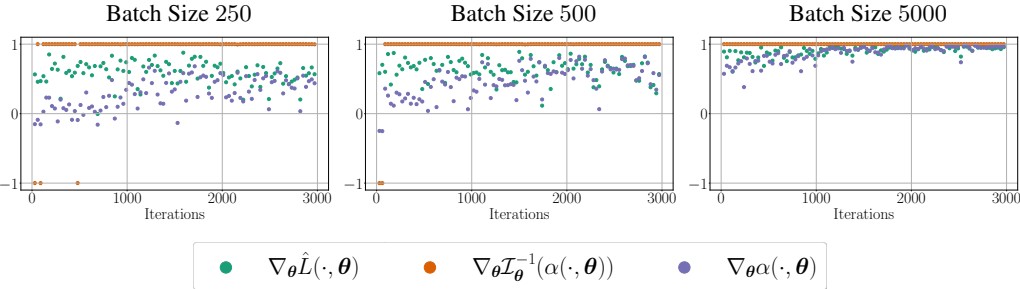

Figure 3: Cosine similarities between using the full dataset $D$ and mini-batches $B_t$ of the three gradient components of Equation (12); namely, train loss, inverse rate and abnormality rate. The same InceptionV3 models of Figure 2 are considered. As shown, gradients of $\mathcal{I}_{\boldsymbol{\theta}_t}^{-1}(\alpha(\cdot,\boldsymbol{\theta}_t))$ are perfectly aligned using $D$ or $B_t$, meaning that batch misalignment in $\nabla_{\boldsymbol{\theta}}\hat{L}(\cdot,\boldsymbol{\theta}_t)$ is governed by $\nabla_{\boldsymbol{\theta}}\alpha(\cdot,\boldsymbol{\theta}_t)$.

deviations, we can use Theorem 4 and compute the probability of observing an $\alpha(D,\boldsymbol{\theta})$ value of 0.7 when the dataset $D$ has a size $n = 50\,000$. This probability is equal to or smaller than $e^{-50\,000 \cdot 0.7} \approx 10^{-8\,000}$, which represents an astronomically small probability. The occurrence of this extremely unlikely event can only be explained by recognizing that gradient descent explores a vast space of different realizations of (potentially independent) random variables $\hat{L}_n(\boldsymbol{\theta})$, one for each model in the model class. When we have a very large model class and explore the empirical loss over a particular dataset $D$, we are really exploring the realizations of a very large number of random variables. It is inevitable that, by chance, some of these realizations will deviate abnormally far from their mean. This phenomenon is independent of the level of concentration of the random variable. The following inequality shows how smaller model classes make $\alpha(D,\boldsymbol{\theta})$ takes smaller values.

**Proposition 5.** *Let be $\boldsymbol{\Theta}$ a finite model class with $M$ models. Then, with h.p. $1-\delta$ over $D \sim \nu^n$,*

$$\mathbb{P}\Big( \bigcap_{\boldsymbol{\theta} \in \boldsymbol{\Theta}} \alpha(D,\boldsymbol{\theta}) \leq \frac{1}{n} \ln \frac{M}{\delta} \Big) \geq 1 - \delta \,.$$

## 4 THE IMPLICIT BIAS OF STOCHASTIC GRADIENT DESCENT (SGD)

Stochastic Gradient Descent (SGD) seeks to minimize $\hat{L}(D,\boldsymbol{\theta})$ by following the gradients of $\hat{L}(B_t,\boldsymbol{\theta}_t)$, where $B_t$ represents the mini-batch processed by SGD at iteration $t$. When batch sizes are large, the gradients of $\hat{L}(D,\boldsymbol{\theta})$ and $\hat{L}(B_t,\boldsymbol{\theta}_t)$ are closely aligned. However, as the batch size decreases, this alignment can deteriorate significantly. This effect is empirically illustrated by the green dots of Figure 3, these green dots display the cosine similarity between $\nabla_{\boldsymbol{\theta}}\hat{L}(B_t,\boldsymbol{\theta}_t)$ and $\nabla_{\boldsymbol{\theta}}\hat{L}(D,\boldsymbol{\theta}_t)$ computed for trained InceptionV3 models with different batch sizes. With large batch sizes, the gradients of $\hat{L}(D,\boldsymbol{\theta}_t)$ and $\hat{L}(B_t,\boldsymbol{\theta}_t)$ are strongly aligned. However, for smaller batch sizes (those typically used in machine learning), the misalignment is much higher.

This misalignment between the gradients of $\hat{L}(D,\boldsymbol{\theta}_t)$ and $\hat{L}(B_t,\boldsymbol{\theta}_t)$ is the effect of the so-called gradient noise introduced by SGD (Keskar et al., 2017; Jastrzebski et al., 2017). This gradient noise is known to be the key factor behind the superior generalization performance of models trained with SGD compared to those trained with GD. Although both SGD and GD converge to neural networks that minimize and interpolate the training data (i.e., $\hat{L}(D,\boldsymbol{\theta}) \approx 0$), the minima found by SGD typically result in better generalization error (Hochreiter and Schmidhuber, 1997). Figure 2 (left) illustrates this widely recognized effect in the literature.

In this section, we show how the misalignment between the gradients of $\hat{L}(D,\boldsymbol{\theta}_t)$ and $\hat{L}(B_t,\boldsymbol{\theta}_t)$ can be clearly identified and understood using the decomposition of the empirical loss presented in Equation (4). Similarly to the gradient decomposition shown in Equation (11), we can decompose the gradient of $\hat{L}(B_t,\boldsymbol{\theta}_t)$ as follows,

$$\nabla_{\boldsymbol{\theta}}\hat{L}(B_t,\boldsymbol{\theta}_t) = \nabla_{\boldsymbol{\theta}}L(\boldsymbol{\theta}_t) - \nabla_{\boldsymbol{\theta}}\mathcal{I}_{\boldsymbol{\theta}_t}^{-1}(s)_{|s=\alpha(B_t,\boldsymbol{\theta}_t)} - \nabla_s\mathcal{I}_{\boldsymbol{\theta}_t}^{-1}(s)_{|s=\alpha(B_t,\boldsymbol{\theta}_t)}\nabla_{\boldsymbol{\theta}}\alpha(B_t,\boldsymbol{\theta}_t)\,. \quad (12)$$

The first component of this decomposition, $\nabla_{\boldsymbol{\theta}}L(\boldsymbol{\theta}_t)$, is independent of the batch $B_t$, and thus remains the same for both GD and SGD. As a result, any differences between the gradients in GD

and SGD must stem from the other two components. However, we will argue that the second term involving $\nabla_{\boldsymbol{\theta}}\mathcal{I}_{\boldsymbol{\theta}_t}^{-1}$ is (nearly) perfectly aligned with the same second component of $\nabla_{\boldsymbol{\theta}}\hat{L}(D, \boldsymbol{\theta})$ beside of the use of mini-batches, concluding that the stochasticity in SGD is governed fully by the last term.

ON THE ALIGNMENT OF $\nabla_{\boldsymbol{\theta}}\mathcal{I}_{\boldsymbol{\theta}_t}^{-1}(s)$ IN GD AND SGD

Figure 3 shows that the cosine similarity between $\nabla_{\boldsymbol{\theta}}\mathcal{I}_{\boldsymbol{\theta}_t}^{-1}(s)_{|s=\alpha(D,\boldsymbol{\theta}_t)}$ and $\nabla_{\boldsymbol{\theta}}\mathcal{I}_{\boldsymbol{\theta}_t}^{-1}(s)_{|s=\alpha(B_t,\boldsymbol{\theta}_t)}$ remains consistently close to 1 or $-1$ across all models encountered during the SGD optimization process, irrespective of the batch size. In fact, this quantity goes to $-1$ only when $L(\boldsymbol{\theta}_t) - \hat{L}(B_t, \boldsymbol{\theta}_t)$ and $L(\boldsymbol{\theta}_t) - \hat{L}(D, \boldsymbol{\theta}_t)$ have different signs, which usually never happens after a few optimization steps, because, after few iterations, we always have that $L(\boldsymbol{\theta}_t) > \hat{L}(B_t, \boldsymbol{\theta}_t)$ and $L(\boldsymbol{\theta}_t) > \hat{L}(D, \boldsymbol{\theta}_t)$, as shown in Figure 2 (left).

We can theoretically explain why the inverse rate gradient of SGD in Equation (12) is perfectly align with its full-batch version at the early stages of the training procedure (when the values of $\alpha(B_t, \boldsymbol{\theta}_t)$ are low) and at the final stages (when $\alpha(B_t, \boldsymbol{\theta}_t)$ is large). Firstly, consider the first iterations of SGD, when $\hat{L}(B_t, \boldsymbol{\theta}_t)$ are still relatively close to $L(\boldsymbol{\theta}_t)$. In that cases, $\alpha(B_t, \boldsymbol{\theta}_t)$ is close to 0, as illustrated in Figure 2 (right), because $\lim_{a \to 0}\mathcal{I}_{\boldsymbol{\theta}}(a) = 0$. In that regime, using a second order approximation around $s = 0$, as shown in Equation (5), we got that $\mathcal{I}_{\boldsymbol{\theta}}^{-1}(s) \approx \text{sign}(s)\sqrt{2|s|}\sigma(\boldsymbol{\theta})$. The direction of the gradient w.r.t. $\boldsymbol{\theta}$ of such quantity does not depend on $s$, and hence, does not depend on $B_t$ through $s = \alpha(B_t, \boldsymbol{\theta}_t)$. As a result, using different mini-batches does not affect the direction of the gradient of $\mathcal{I}_{\boldsymbol{\theta}}^{-1}(s)$ at the early stages of the training setup. The particular mini-batch only affects the norm of this gradient. On the other hand, at the latter stages of the learning when $\hat{L}(B_t, \boldsymbol{\theta}_t) \approx 0$, by adapting Proposition 3 for batches $B_t$, we have that $\mathcal{I}_{\boldsymbol{\theta}}^{-1}(\alpha(B_t, \boldsymbol{\theta}_t)) \approx L(\boldsymbol{\theta}_t)$ and the same argument holds: the different mini-batches does not affect direction of the gradient $\nabla_{\boldsymbol{\theta}}\mathcal{I}_{\boldsymbol{\theta}}^{-1}(s)_{|s=\alpha(B_t,\boldsymbol{\theta}_t)}$. These approximations are shown in Figure A.6 for different batch sizes. They hold quite well for a large part of the training process.

Our hypothesis to explain the perfect alignment observed in the middle phase of training is that $\nabla_{\boldsymbol{\theta}}\mathcal{I}_{\boldsymbol{\theta}}^{-1}(s)$ can also be accurately approximated as the product of two functions, $\nabla_{\boldsymbol{\theta}}\mathcal{I}_{\boldsymbol{\theta}}^{-1}(s) = f(s, \boldsymbol{\theta})\nabla_{\boldsymbol{\theta}}g(\boldsymbol{\theta})$, similar to what occurs in the early and late training stages. As a result, the abnormality rate $\alpha(B_t, \boldsymbol{\theta}_t)$ influences only the magnitude, not the direction, of the inverse rate's gradient. Although a exploration of this decoupled gradient approximation is beyond the scope of this work, the following result demonstrates that, for linearized neural networks (a commonly used approximation valid in the infinite-width limit (Jacot et al., 2018)) and under certain assumptions about the data-generating distribution, this gradient decoupling of the inverse rate indeed always holds.

**Proposition 6** (Informal). *In regression problems with mean squared error loss and under a model linearization hypothesis with Gaussian feature vectors, the gradient of the inverse-rate function can be expressed as $\nabla_{\boldsymbol{\theta}}\mathcal{I}_{\boldsymbol{\theta}}^{-1}(s) = f(s, \boldsymbol{\theta})\nabla_{\boldsymbol{\theta}}\sigma(\boldsymbol{\theta})$.*

The conclusion of all these analyses is that both the first and second components of the stochastic gradient $\nabla_{\boldsymbol{\theta}}\hat{L}(B_t, \boldsymbol{\theta}_t)$ are perfectly aligned with the corresponding components of the full-batch gradient $\nabla_{\boldsymbol{\theta}}\hat{L}(D, \boldsymbol{\theta}_t)$. Therefore, the primary source of gradient noise in SGD arises from the misalignment between the third components of the stochastic and full-batch gradients.

SGD PREVENTS HIGHLY ABNORMAL GENERALIZATION ERRORS

SGD promotes models with abnormal generalization errors due to $\nabla_{\boldsymbol{\theta}}\alpha(D, \boldsymbol{\theta}_t)$ appearing in the decomposition of Equation (12). Figure 3 shows how, under large batches, $\nabla_{\boldsymbol{\theta}}\alpha(B_t, \boldsymbol{\theta}_t)$ is almost perfectly aligned with $\nabla_{\boldsymbol{\theta}}\alpha(D, \boldsymbol{\theta}_t)$, directing the optimizer towards models with abnormal generalization errors (i.e., models with larger $\alpha(D, \boldsymbol{\theta}_t)$ values) as shown in Figure 2 (right). However, this figure also shows how SGD with small mini-batches leads to models with much less abnormal generalization errors. In this case, Figure 3 (left) shows how the the gradients $\nabla_{\boldsymbol{\theta}}\alpha(B_t, \boldsymbol{\theta}_t)$ are highly misaligned with $\nabla_{\boldsymbol{\theta}}\alpha(D, \boldsymbol{\theta}_t)$. It is important to observe that, in the gradient decompositions of GD and SGD given by Equations (11) and (12), the final term is multiplied by a gradient, which acts as a scalar. Consequently, the cosine distance between these third components is equivalent to that between the $\alpha(D, \boldsymbol{\theta}_t)$ and $\alpha(B_t, \boldsymbol{\theta}_t))$ components.

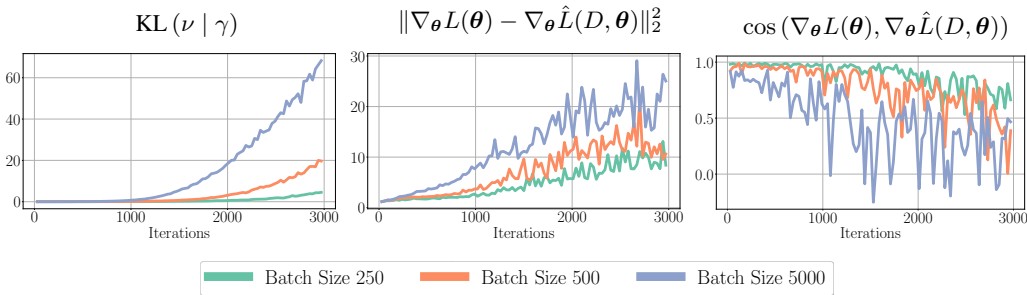

Figure 4: Evolution on KL divergence from Theorem 7 (left), norm difference between training loss and expected loss gradients (center) and cosine similarity between the gradients (right) of the InceptionV3 models trained on Cifar10 for different batch sizes.

Let denote $\boldsymbol{\theta}_t^\times$ an update of $\boldsymbol{\theta}_t$ by following the gradient of $\alpha(B_t, \boldsymbol{\theta}_t)$ instead of $\alpha(D, \boldsymbol{\theta}_t)$. That is, $\boldsymbol{\theta}_t^\times = \boldsymbol{\theta}_t + \gamma \nabla_{\boldsymbol{\theta}} \alpha(B_t, \boldsymbol{\theta}_t)$ for a step-size $\gamma > 0$. The alignment between $\nabla_{\boldsymbol{\theta}} \alpha(B_t, \boldsymbol{\theta}_t)$ and $\nabla_{\boldsymbol{\theta}} \alpha(D, \boldsymbol{\theta}_t)$ determines the value of $\alpha(D, \boldsymbol{\theta}_t^\times)$. When $\gamma$ is small, we can use a Taylor approximation of order 1 on $\alpha(D, \boldsymbol{\theta}_t^\times)$, centered at $\boldsymbol{\theta}_t$, to estimate $\alpha(D, \boldsymbol{\theta}_t^\times)$,

$$\alpha(D, \boldsymbol{\theta}_t^\times) \approx \alpha(D, \boldsymbol{\theta}_t) + \gamma \nabla_{\boldsymbol{\theta}} \alpha(D, \boldsymbol{\theta}_t)^T \nabla_{\boldsymbol{\theta}} \alpha(B_t, \boldsymbol{\theta}_t).$$

Naming $\beta$ the angle between $\nabla_{\boldsymbol{\theta}} \alpha(D, \boldsymbol{\theta}_t)$ and $\nabla_{\boldsymbol{\theta}} \alpha(B_t, \boldsymbol{\theta}_t)$, that is, their *cosine similarity*, we can rewrite the above equation as

$$\alpha(D, \boldsymbol{\theta}_t^\times) \approx \alpha(D, \boldsymbol{\theta}_t) + \gamma \|\nabla_{\boldsymbol{\theta}} \alpha(D, \boldsymbol{\theta}_t)\| \|\nabla_{\boldsymbol{\theta}} \alpha(B_t, \boldsymbol{\theta}_t)\| \cos(\beta).$$

As a result, if the two gradients are highly misaligned, $\cos(\beta)$ will take on small positive or even negative values, as illustrated in Figure 3 (left). This leads to smaller *increases* or even *decreases* in $\alpha(D, \boldsymbol{\theta}_t)$. In contrast, as shown in Figure 3 (right), using larger batches results in $\cos(\beta)$ values that are closer to 1, which facilitates a more straightforward increase in $\alpha(D, \boldsymbol{\theta}_t)$.

Overall, we observe that the primary effect of gradient noise in SGD is to bias the optimizer toward models with less abnormal generalization errors, which, according to Proposition 3, leads to models with smaller generalization error. In the next section, we explore another consequence of this bias.

ON WHY SGD IS BIASED TOWARDS MODELS WITH LOWER GENERALIZATION ERROR

On average, SGD follows the gradients of $\hat{L}(D, \boldsymbol{\theta})$, meaning $\mathbb{E}_{B \sim D}[\nabla_{\boldsymbol{\theta}} L(B, \boldsymbol{\theta})] = \nabla_{\boldsymbol{\theta}} \hat{L}(D, \boldsymbol{\theta})$, where $B \sim D$ represents the mini-batches sampled from the dataset $D$. In the next result, we show that the similarity between the gradients of $\hat{L}(D, \boldsymbol{\theta})$ and $L(\boldsymbol{\theta})$ improves for models whose generalization error is less abnormal. The key result is that the gradient of $\hat{L}(D, \boldsymbol{\theta})$ can be represented as an expectation over an alternative distribution $\gamma$, where the KL divergence between $\gamma$ and the true data distribution $\nu$ increases with the abnormality $\alpha(D, \boldsymbol{\theta})$. By keeping this abnormality low, the gradients of $\hat{L}(D, \boldsymbol{\theta})$ for the models visited by SGD are more similar to those of $L(\boldsymbol{\theta})$, leading to models with lower expected loss.

**Theorem 7.** *For any $\boldsymbol{\theta} \in \Theta$, $n > 0$ and $D \sim \nu^n$, there exists a distribution $\gamma(\boldsymbol{y}, \boldsymbol{x})$ that depends on $\boldsymbol{\theta}$ and $\alpha(D, \boldsymbol{\theta})$, such that,*

$$\nabla_{\boldsymbol{\theta}} \hat{L}(D, \boldsymbol{\theta}) = \nabla_{\boldsymbol{\theta}} \mathbb{E}_\gamma[\ell(\boldsymbol{y}, \boldsymbol{x}, \boldsymbol{\theta})] \tag{13}$$

*and $KL(\nu \mid \gamma)$ is monotonically increasing with $\alpha(D, \boldsymbol{\theta})$ and $KL(\nu \mid \gamma) = 0$ if $\alpha(D, \boldsymbol{\theta}) = 0$. Furthermore, if the loss function $\ell(\boldsymbol{y}, \boldsymbol{x}, \boldsymbol{\theta})$ is $M$-Lipschitz with respect to $(\boldsymbol{y}, \boldsymbol{x})$. Then,*

$$\|\nabla_{\boldsymbol{\theta}} \mathcal{I}_{\boldsymbol{\theta}}^{-1}(s)_{|s = \alpha(D, \boldsymbol{\theta})}\|_2 \le M \sqrt{2\,KL(\nu \mid \gamma)}. \tag{14}$$

Equation (13) suggests that the difference between the expected and empirical gradients is thus governed by the *difference* between $\nu$ and $\gamma$,

$$\nabla L(\boldsymbol{\theta}) = \nabla_{\boldsymbol{\theta}} E_\nu[\ell(\boldsymbol{x}, \boldsymbol{y}, \boldsymbol{\theta})], \qquad \nabla \hat{L}(D, \boldsymbol{\theta}) = \nabla_{\boldsymbol{\theta}} E_\gamma[\ell(\boldsymbol{x}, \boldsymbol{y}, \boldsymbol{\theta})].$$

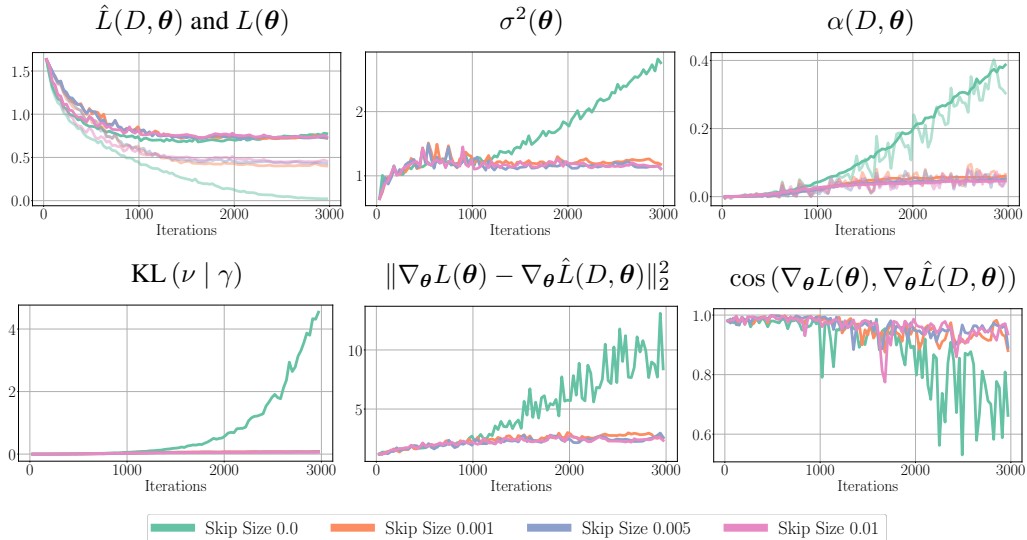

Figure 5: Evolution on train/test loss (upper left), variance (upper center), abnormality rate (upper right), KL divergence from Theorem 7 (lower left), distance (lower center) and cosine similarity (lower right) between training loss and expected loss gradients of the InceptionV3 models trained on Cifar10 using batch size of 250 and the skipping procedure described in Section 4.

Following the procedures in Masegosa and Ortega (2024), $\mathrm{KL}\left(\nu \mid \gamma\right)$ can be easily estimated using Proposition 16 and the test set. Figure 4 (left) illustrates that SGD with smaller mini-batches tends to explore models where this KL divergence is significantly reduced, which, as shown in Theorem 7, results from the decreased level of abnormality $\alpha(D, \boldsymbol{\theta})$ in the models visited by SGD. Consequently, the gradients of $\hat{L}(D, \boldsymbol{\theta})$ and $L(\boldsymbol{\theta})$ should become more similar, as suggested by Theorem 7. The experimental findings presented in Figure 4 (center and right) support this theoretical analysis.

Equation (14) in Theorem 7 shows how the $\mathrm{KL}\left(\nu \mid \gamma\right)$ term also limits the norm of the second component of the gradient of $\hat{L}(D, \boldsymbol{\theta})$ in Equation (11), which also aligns with the idea that the gradient of $\hat{L}(D, \boldsymbol{\theta})$ become more similar to the gradient of $L(\boldsymbol{\theta})$ according the decomposition given in Equation (11). As a consequence, a smaller abnormality induces a smaller KL term and, in turn, a gradient of the inverse rate with smaller norm. The consequence is that the optimizer is less biased to models with a poorly concentrated loss. Figure A.7 (left) shows how the norm of this gradient is (relatively) smaller for SGD with smaller mini-batches. And this would explain why SGD is also biased towards models with more concentrated losses, as shown in Figure 2 (center).

DISCARDING HIGHLY ABNORMAL MINI-BATCHES

To further validate our results, we conducted an experiment in Figure 5 where we applied SGD optimization but discarded batches with $\alpha(B_t, \boldsymbol{\theta}_t)$ values deemed *large*. We used Theorem 4 to determine when an $\alpha(B_t, \boldsymbol{\theta}_t)$ value was considered *large*, discarding batches where $\alpha(B_t, \boldsymbol{\theta}_t)$ exceeded a pre-specified quantile of the corresponding exponential distribution. The rationale is that if a batch is highly abnormal (i.e., the probability of observing it is below 0.001, this threshold is called *skip size* in Figure 5), the similarity between its gradient and the gradient of the expected loss is likely to be poor, as the term $\mathrm{KL}\left(\nu \mid \gamma\right)$, which controls this similarity, would be large according to Theorem 7. Therefore, in such cases, it's more effective to skip the batch, avoid following its gradient, wait for the next batch and repeats the procedure.

Figure 5 supports our theoretical analysis. The reduction in the level of abnormality leads to effects analogous to those seen when reducing the batch size, as when transitioning from GD to SGD. This corroborates the idea that by reducing the level of abnormality, we can biased the optimizer towards models with smaller generalization error. This experiment should not be interpreted as a novel training approach because to compute $\alpha(B_t, \boldsymbol{\theta}_t)$, we are using the test set to approximate $\nu$.

## 5 RELATED WORK

The generalization capabilities of Stochastic Gradient Descent (SGD) have been extensively studied, with various theories proposed to explain why SGD often outperforms deterministic optimization methods in terms of generalization. A prominent line of research attributes this phenomenon to the tendency of SGD to converge to flat minima in the loss landscape. Hochreiter and Schmidhuber (1997) first introduced the concept of flat minima, suggesting that solutions located in wide, flat regions of the loss surface generalize better to unseen data. This idea has been further explored by many other works (Keskar et al., 2016), who observed that small-batch SGD gravitates towards flatter minima, while large-batch training tends to find sharp minima associated with poorer generalization. Although our work focuses on the concentration properties of the empirical loss rather than the geometry of the loss landscape, our main hypothesis is that the connection between these two lines of research is that flatter minima correspond to models which are more concentrated and/or with less abnormal generalization error.

Another perspective considers the implicit regularization effect of SGD. Neyshabur et al. (2015a) proposed that SGD biases models towards solutions with smaller norms, aligning with capacity control theories that relate model complexity to generalization. This implicit norm minimization effect has been linked to the generalization performance of deep neural networks, as networks with smaller weights are thought to be less prone to overfitting (Bartlett et al., 2017). The work of Masegosa and Ortega (2024) would establish a link between these works and our work, as it establishes that models with smaller norms exhibit greater concentration in their empirical losses.

Works using concentration bounds to understand SGD build on the same conceptualization by treating the empirical loss of each model as a random variable (Kawaguchi et al., 2017; Bartlett et al., 2017; Neyshabur et al., 2017; Golowich et al., 2018; Liang et al., 2019). However, these works typically rely on upper bounds, which are often known to be vacuous or overly loose in deep neural networks (Nagarajan and Kolter, 2019; Gastpar et al., 2024). Moreover, they generally do not account for the individual concentration properties of each model in the hypothesis space, potentially overlooking critical nuances in how different models generalize (Casado et al., 2024). In contrast, our work leverages a fundamental equality that directly decomposes the training loss into distinct components, providing a more nuanced and detailed analysis that goes beyond the limitations of traditional concentration bounds.

## 6 CONCLUSIONS AND LIMITATIONS

In this work, we have presented a novel theoretical analysis of Stochastic Gradient Descent (SGD) using principles from Large Deviation Theory (LDT). Our findings reveal that the generalization error in SGD can be decomposed into components influenced by the expected loss, the concentration of the empirical loss, and the level of abnormal deviations from the expected value.

Our analysis reveals that the primary effect of gradient noise in SGD is to limit the exploration of models where the empirical loss deviates substantially from the expected loss. We show that this effect ensures that SGD tends to visit models where the empirical gradients closely align with the expected gradients, resulting in a more effective reduction of the expected loss and, consequently, leading to models with lower generalization error.

While this work offers valuable theoretical insights into the implicit regularization effects of SGD, there are several limitations that need to be addressed. Firstly, although we introduced the concept of SKIP-SGD, we did not provide a fully developed alternative to standard SGD. We believe, however, that this approach could be transformed into a viable optimization technique by using an independent validation dataset to compute $\alpha(B_t, \boldsymbol{\theta}_t)$, potentially paving the way for entirely new variations of SGD. Secondly, our empirical findings were limited to a specific set of models, and it is crucial to validate these results across a broader range of architectures and tasks to ensure their generalizability. Lastly, the role of explicit regularization methods, such as $\ell_2$ regularization or the use of invariant architectures, should be investigated within this framework to better understand how they interact with and influence the implicit biases of SGD.

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

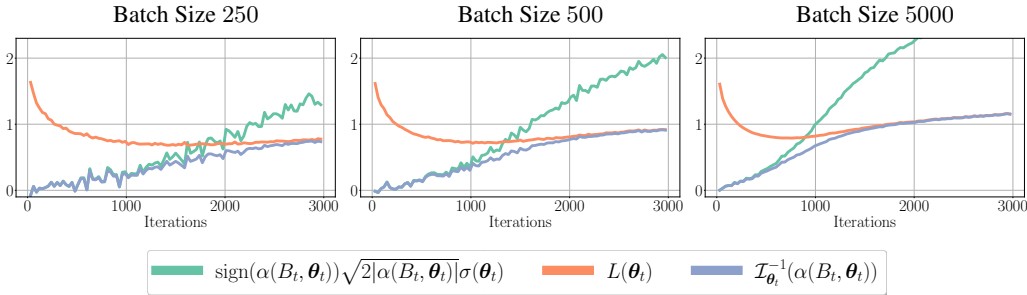

Figure A.6: Evolution of the inverse rate evaluated at $\alpha(B_t, \boldsymbol{\theta}_t)$. Two other functions are also shown, one that perfectly fits the inverse rate at the early stages and another that perfectly fits for the latter stages of the training procedure. The same InceptionV3 models of Figure 2 are considered.

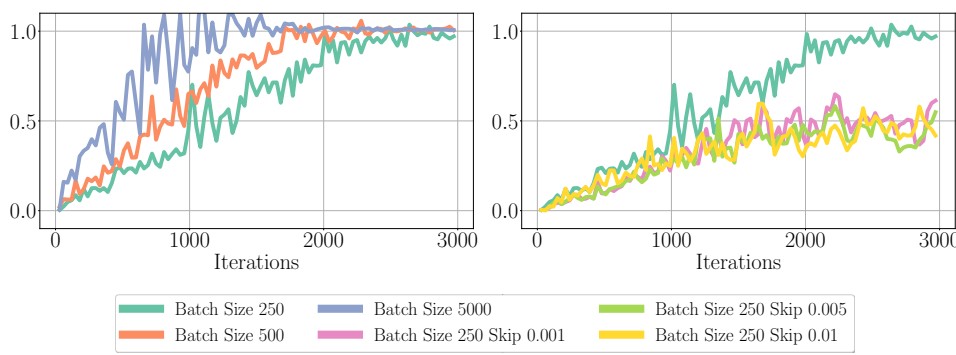

Figure A.7: Evolution of $\|\nabla_{\boldsymbol{\theta}} \mathcal{I}_{\boldsymbol{\theta}_t}^{-1}(s)_{|s=\alpha(D,\boldsymbol{\theta}_t)}\|_2$ divided by $\|\nabla_{\boldsymbol{\theta}} L(\boldsymbol{\theta}_t)\|_2$ for different InceptionV3 models trained with different batch sizes and SGD-SKIP procedures. The same models that were used in Figures 2 and 5 are considered here.

## A   EXPERIMENTAL DETAILS

The conducted experimentation can be found in the anonymous Github Repository https://github.com/SGDAbnormality/SGDAbnormality.

## B   THEOREMS AND PROOFS

**Proposition 3.** For any $D \sim \nu^n$ and any $\boldsymbol{\theta} \in \boldsymbol{\Theta}$, we have that

$$\hat{L}(D, \boldsymbol{\theta}) = L(\boldsymbol{\theta}) - \mathcal{I}_{\boldsymbol{\theta}}^{-1}\left(\alpha(D, \boldsymbol{\theta})\right) . \tag{15}$$

where $\alpha : \mathcal{D} \times \boldsymbol{\Theta} \to \mathbb{R}$ is defined as $\alpha(D, \boldsymbol{\theta}) := \mathcal{I}_{\boldsymbol{\theta}}(L(\boldsymbol{\theta}) - \hat{L}(D, \boldsymbol{\theta}))$.

*Proof.* This is a direct consequence of the fact that the (signed) rate function $\mathcal{I}_{\boldsymbol{\theta}}(\cdot)$ is a bijective function. This in turn is a consequence of the fact that the non-signed rate function is strictly convex and positive in $\mathbb{R}$ (Rockafellar, 1970). $\square$

**Theorem 4.** For any $\boldsymbol{\theta} \in \boldsymbol{\Theta}$, $n > 0$ and $D \sim \nu^n$, the cumulative of distribution of $\alpha(D, \boldsymbol{\theta})$ satisfies

$$\forall s > 0 \quad \mathbb{P}_{D \sim \nu^n}\left(\alpha(D, \boldsymbol{\theta}) \geq s\right) \leq e^{-n|s|} \quad \text{and} \quad \forall s < 0 \quad \mathbb{P}_{D \sim \nu^n}\left(\alpha(D, \boldsymbol{\theta}) \leq s\right) \leq e^{-n|s|}, \tag{16}$$

and both inequalities are asymptotically tight,

$$\forall s > 0 \quad \mathbb{P}_{D \sim \nu^n}(\alpha(D, \boldsymbol{\theta}) \geq s) \asymp e^{-n|s|} \quad \text{and} \quad \forall s < 0 \quad \mathbb{P}_{D \sim \nu^n}(\alpha(D, \boldsymbol{\theta}) \leq s) \asymp e^{-n|s|} . \tag{17}$$

*Proof.* From Theorem 1 we got that

$$\forall a \geq 0, \quad \mathbb{P}_{D \sim \nu^n}\big(L(\boldsymbol{\theta}) - \hat{L}(D, \boldsymbol{\theta}) \geq a\big) \leq e^{-n|\mathcal{I}_{\boldsymbol{\theta}}(a)|},$$
$$\forall a \leq 0, \quad \mathbb{P}_{D \sim \nu^n}\big(L(\boldsymbol{\theta}) - \hat{L}(D, \boldsymbol{\theta}) \leq a\big) \leq e^{-n|\mathcal{I}_{\boldsymbol{\theta}}(a)|}. \tag{18}$$

As a result, for any value of $s \in \mathbb{R}$, taking $a = \mathcal{I}_{\boldsymbol{\theta}}^{-1}(s)$, we got

$$\forall s \geq 0, \quad \mathbb{P}_{D \sim \nu^n}\big(L(\boldsymbol{\theta}) - \hat{L}(D, \boldsymbol{\theta}) \geq \mathcal{I}_{\boldsymbol{\theta}}^{-1}(s)\big) \leq e^{-n|s|},$$
$$\forall s \leq 0, \quad \mathbb{P}_{D \sim \nu^n}\big(L(\boldsymbol{\theta}) - \hat{L}(D, \boldsymbol{\theta}) \leq \mathcal{I}_{\boldsymbol{\theta}}^{-1}(s)\big) \leq e^{-n|s|}. \tag{19}$$

As $\mathcal{I}_{\boldsymbol{\theta}}(\cdot)$ is a strictly monotonic and increasing function, we an apply it at both sides of the inequality inside the probability, giving as:

$$\forall s \geq 0, \quad \mathbb{P}_{D \sim \nu^n}\big(\mathcal{I}_{\boldsymbol{\theta}}(L(\boldsymbol{\theta}) - \hat{L}(D, \boldsymbol{\theta})) \geq s\big) \leq e^{-n|s|},$$
$$\forall s \leq 0, \quad \mathbb{P}_{D \sim \nu^n}\big(\mathcal{I}_{\boldsymbol{\theta}}(L(\boldsymbol{\theta}) - \hat{L}(D, \boldsymbol{\theta})) \leq s\big) \leq e^{-n|s|}. \tag{20}$$

The asymptotic inequalities can be obtained by applying the same reasoning to Equation (3), which is a direct consequence of Cramér's Theorem. □

**Proposition 5.** *Let be $\Theta$ a finite model class with $M$ models. Then, with h.p. $1 - \delta$ over $D \sim \nu^n$,*

$$\mathbb{P}\Big(\bigcap_{\boldsymbol{\theta} \in \Theta} \alpha(D, \boldsymbol{\theta}) \leq \frac{1}{n} \ln \frac{M}{\delta}\Big) \geq 1 - \delta.$$

*Proof.* By Chernoff's Theorem 1, for a given $\boldsymbol{\theta}$, we have, that for any $a \geq 0$, it verifies that $\mathbb{P}(L(\boldsymbol{\theta}) - \hat{L}(D, \boldsymbol{\theta}) \geq a) \leq e^{-n|\mathcal{I}_{\boldsymbol{\theta}}(a)|}$. Naming $\delta' := e^{-n|\mathcal{I}_{\boldsymbol{\theta}}(a)|} \leq 1$ and re-arranging terms, $a = \mathcal{I}_{\boldsymbol{\theta}}^{-1}\big(-\frac{1}{n} \ln \delta'\big) \geq 0$. This allows us to rewrite the first equation as

$$\mathbb{P}\Big(L(\boldsymbol{\theta}) - \hat{L}(D, \boldsymbol{\theta}) \geq \mathcal{I}_{\boldsymbol{\theta}}^{-1}\big(\tfrac{1}{n} \ln \tfrac{1}{\delta'}\big)\Big) \leq \delta'.$$

Using that the rate function $\mathcal{I}_{\boldsymbol{\theta}}(\cdot)$ is a bijection, we got that

$$\mathbb{P}\Big(\mathcal{I}_{\boldsymbol{\theta}}(L(\boldsymbol{\theta}) - \hat{L}(D, \boldsymbol{\theta})) \geq \tfrac{1}{n} \ln \tfrac{1}{\delta'}\Big) \leq \delta' \implies \mathbb{P}\Big(\alpha(D, \boldsymbol{\theta}) \geq \tfrac{1}{n} \ln \tfrac{1}{\delta'}\Big) \leq \delta'.$$

Using an union bound over the set of $M$ models,

$$\mathbb{P}\Big(\bigcup_{\boldsymbol{\theta} \in \Theta} \alpha(D, \boldsymbol{\theta}) \geq \tfrac{1}{n} \ln \tfrac{1}{\delta'}\Big) \leq \sum_{\boldsymbol{\theta} \in \Theta} \mathbb{P}\Big(\alpha(D, \boldsymbol{\theta}) \geq \tfrac{1}{n} \ln \tfrac{1}{\delta'}\Big).$$

As we have $M$ different models, the r.h.s. can be rewritten as

$$\mathbb{P}\Big(\bigcup_{\boldsymbol{\theta} \in \Theta} \alpha(D, \boldsymbol{\theta}) \geq \tfrac{1}{n} \ln \tfrac{1}{\delta'}\Big) \leq M\delta'.$$

By reparametrizing the above inequality with $\delta' = \delta M^{-1}$ we have

$$\mathbb{P}\Big(\bigcup_{\boldsymbol{\theta} \in \Theta} L(\boldsymbol{\theta}) - \hat{L}(D, \boldsymbol{\theta}) \geq \frac{1}{n} \ln \frac{M}{\delta}\Big) \leq \delta.$$

Which verifies,

$$1 - \mathbb{P}\Big(\bigcup_{\boldsymbol{\theta} \in \Theta} \alpha(D, \boldsymbol{\theta}) \geq \frac{1}{n} \ln \frac{M}{\delta}\Big) \geq 1 - \delta.$$

Which is equivalent to,

$$\mathbb{P}\Big(\bigcap_{\boldsymbol{\theta} \in \Theta} \alpha(D, \boldsymbol{\theta}) \leq \frac{1}{n} \ln \frac{M}{\delta}\Big) \geq 1 - \delta.$$

□

**Proposition 8.** *Let $\mathcal{A}, \mathcal{B}$ be open sets and $f : \mathcal{A} \times \mathcal{B} \to \mathbb{R}$ be a function on $\mathbb{R}$. If we denote $b_a^\star$ the maximum or the minimum over $\mathcal{B}$ for a fixed $a \in \mathcal{A}$, i.e.,*

$$b_a^\star = \arg\max_b f(a, b) \quad or \quad b_a^\star = \arg\min_b f(a, b) \tag{21}$$

*Then, we have that*

$$\nabla_a f(a, b_a^\star) = \nabla_a f(a, b)_{|b = b_a^\star} \tag{22}$$

*Proof.* It is clear that, using the chain rule

$$\nabla_a f(a, b_a^\star) = \nabla_a f(a, b)_{|b=b_a^\star} + \nabla_b f(a, b)_{b=b_a^\star} \nabla_a b_a^\star . \tag{23}$$

However, given that $b_a^\star$ is an optimal value, it verifies that $\nabla_b f(a, b)_{b=b_a^\star} = 0$ by definition of maximum/minimum. As a result,

$$\nabla_a f(a, b_a^\star) = \nabla_a f(a, b)_{|b=b_a^\star} . \tag{24}$$

$\square$

**Proposition 9.** *For any model $\boldsymbol{\theta} \in \Theta$, it verifies that*

$$\forall s > 0 \quad \nabla_{\boldsymbol{\theta}} \mathcal{I}_{\boldsymbol{\theta}}^{-1}(s) = \frac{1}{\lambda^s} \nabla_{\boldsymbol{\theta}} J_{\boldsymbol{\theta}}(\lambda^s) , \tag{25}$$

*where $\lambda^s$ is defined as,*

$$\lambda^s = \arg\min_{\lambda > 0} \frac{J_{\boldsymbol{\theta}}(\lambda) + s}{\lambda} . \tag{26}$$

*Proof.* Given that tha minimum in the refinition of the inverse rate function is reached, we can express $\mathcal{I}_{\boldsymbol{\theta}}^{-1}(a)$ as follows,

$$\mathcal{I}_{\boldsymbol{\theta}}^{-1}(s) = \frac{J_{\boldsymbol{\theta}}(\lambda^s) + s}{\lambda^s} . \tag{27}$$

Then $\nabla_{\boldsymbol{\theta}} \mathcal{I}_{\boldsymbol{\theta}}^{-1}(s)$ can be computed as

$$\nabla_{\boldsymbol{\theta}} \mathcal{I}_{\boldsymbol{\theta}}^{-1}(s) = \nabla_{\boldsymbol{\theta}} \frac{J_{\boldsymbol{\theta}}(\lambda^s) + s}{\lambda^s} . \tag{28}$$

And, by Proposition 8, this gradient does not have to propagate through $\lambda^s$. Then, it simplifies to,

$$\nabla_{\boldsymbol{\theta}} \mathcal{I}_{\boldsymbol{\theta}}^{-1}(s) = \frac{1}{\lambda^s} \nabla_{\boldsymbol{\theta}} J_{\boldsymbol{\theta}}(\lambda^s) . \tag{29}$$

which concludes the proof. $\square$

**Proposition 10.** *Under the setup where the loss function is*

$$\ell(y, \boldsymbol{x}, \boldsymbol{\theta}) = (y - f_{\boldsymbol{\theta}}(\boldsymbol{x}))^2 ,$$

*with $f_{\boldsymbol{\theta}}(\boldsymbol{x})$ linearized around $\boldsymbol{\theta}_0$:*

$$f_{\boldsymbol{\theta}}(\boldsymbol{x}) \approx f_{\boldsymbol{\theta}_0}(\boldsymbol{x}) + \nabla_{\boldsymbol{\theta}} f_{\boldsymbol{\theta}_0}(\boldsymbol{x})^\top (\boldsymbol{\theta} - \boldsymbol{\theta}_0) ,$$

*and assuming that*

$$y = f_{\boldsymbol{\theta}_0}(\boldsymbol{x}) + \nabla_{\boldsymbol{\theta}} f_{\boldsymbol{\theta}_0}(\boldsymbol{x})^\top (\boldsymbol{\theta}^\star - \boldsymbol{\theta}_0) ,$$

*with $\nabla_{\boldsymbol{\theta}} f_{\boldsymbol{\theta}_0}(\boldsymbol{x}) \sim \mathcal{N}(\boldsymbol{0}, \Sigma)$, the variance of the loss function $\ell(y, \boldsymbol{x}, \boldsymbol{\theta})$ under the distribution $\nu(y, \boldsymbol{x})$ is*

$$\mathrm{Var}_\nu \left[ \ell(y, \boldsymbol{x}, \boldsymbol{\theta}) \right] = 2 \left( L(\boldsymbol{\theta}) \right)^2 ,$$

*where $L(\boldsymbol{\theta}) = (\boldsymbol{\theta}^\star - \boldsymbol{\theta})^\top \Sigma (\boldsymbol{\theta}^\star - \boldsymbol{\theta})$ is the expected loss.*

*Proof.* From the given assumptions, the loss function simplifies to

$$\ell(y, \boldsymbol{x}, \boldsymbol{\theta}) = \left( \nabla_{\boldsymbol{\theta}} f_{\boldsymbol{\theta}_0}(\boldsymbol{x})^\top (\boldsymbol{\theta}^\star - \boldsymbol{\theta}) \right)^2 .$$

Let $\delta\boldsymbol{\theta} = \boldsymbol{\theta}^\star - \boldsymbol{\theta}$ and define the random variable

$$Z = \nabla_{\boldsymbol{\theta}} f_{\boldsymbol{\theta}_0}(\boldsymbol{x})^\top \delta\boldsymbol{\theta} .$$

Since $\nabla_{\boldsymbol{\theta}} f_{\boldsymbol{\theta}_0}(\boldsymbol{x}) \sim \mathcal{N}(\boldsymbol{0}, \Sigma)$, it follows that

$$Z \sim \mathcal{N}\left(0, \sigma^2\right) ,$$

where

$$\sigma^2 = \delta\boldsymbol{\theta}^\top \Sigma \delta\boldsymbol{\theta} = L(\boldsymbol{\theta}) .$$

Therefore, the loss function can be expressed as

$$\ell(y, \boldsymbol{x}, \boldsymbol{\theta}) = Z^2 .$$

To find the variance of $\ell(y, \boldsymbol{x}, \boldsymbol{\theta})$, we compute

$$\text{Var}_\nu \left[ \ell(y, \boldsymbol{x}, \boldsymbol{\theta}) \right] = \mathbb{E}_\nu \left[ (\ell(y, \boldsymbol{x}, \boldsymbol{\theta}))^2 \right] - \left( \mathbb{E}_\nu \left[ \ell(y, \boldsymbol{x}, \boldsymbol{\theta}) \right] \right)^2 .$$

First, compute the expected value:

$$\mathbb{E}_\nu \left[ \ell(y, \boldsymbol{x}, \boldsymbol{\theta}) \right] = \mathbb{E}_\nu \left[ Z^2 \right] = \sigma^2 = L(\boldsymbol{\theta}) .$$

Next, compute the fourth moment:

$$\mathbb{E}_\nu \left[ (\ell(y, \boldsymbol{x}, \boldsymbol{\theta}))^2 \right] = \mathbb{E}_\nu \left[ Z^4 \right] .$$

Since $Z$ is normally distributed with mean zero and variance $\sigma^2$, the fourth moment is

$$\mathbb{E}_\nu \left[ Z^4 \right] = 3\sigma^4 .$$

Therefore, the variance is

$$\begin{aligned}
\text{Var}_\nu \left[ \ell(y, \boldsymbol{x}, \boldsymbol{\theta}) \right] &= \mathbb{E}_\nu \left[ Z^4 \right] - \left( \mathbb{E}_\nu \left[ Z^2 \right] \right)^2 \\
&= 3\sigma^4 - (\sigma^2)^2 \\
&= 3\sigma^4 - \sigma^4 \\
&= 2\sigma^4 \\
&= 2 \left( L(\boldsymbol{\theta}) \right)^2 .
\end{aligned}$$

This completes the proof. $\qquad\square$

**Proposition 11.** *Consider a regression problem with the mean squared error loss function*

$$\ell(y, \boldsymbol{x}, \boldsymbol{\theta}) = (y - f_{\boldsymbol{\theta}}(\boldsymbol{x}))^2 ,$$

*where $f_{\boldsymbol{\theta}}(\boldsymbol{x})$ is approximated by a first-order Taylor expansion around an initial parameter $\boldsymbol{\theta}_0$:*

$$f_{\boldsymbol{\theta}}(\boldsymbol{x}) \approx f_{\boldsymbol{\theta}_0}(\boldsymbol{x}) + \nabla_{\boldsymbol{\theta}} f_{\boldsymbol{\theta}_0}(\boldsymbol{x})^\top (\boldsymbol{\theta} - \boldsymbol{\theta}_0) .$$

*Assume that the target variable $y$ is given by*

$$y = f_{\boldsymbol{\theta}_0}(\boldsymbol{x}) + \nabla_{\boldsymbol{\theta}} f_{\boldsymbol{\theta}_0}(\boldsymbol{x})^\top (\boldsymbol{\theta}^\star - \boldsymbol{\theta}_0) ,$$

*for some parameter $\boldsymbol{\theta}^\star \in \boldsymbol{\Theta}$, and that the gradients $\nabla_{\boldsymbol{\theta}} f_{\boldsymbol{\theta}_0}(\boldsymbol{x})$ follow a multivariate normal distribution:*

$$\nabla_{\boldsymbol{\theta}} f_{\boldsymbol{\theta}_0}(\boldsymbol{x}) \sim \mathcal{N}(\boldsymbol{0}, \Sigma) .$$

*Then, the cumulant generating function $J_{\boldsymbol{\theta}}(\lambda)$ of the centered loss $L(\boldsymbol{\theta}) - \ell(y, \boldsymbol{x}, \boldsymbol{\theta})$ is given by*

$$J_{\boldsymbol{\theta}}(\lambda) = \lambda L(\boldsymbol{\theta}) - \frac{1}{2} \ln \left( 1 + 2\lambda L(\boldsymbol{\theta}) \right) ,$$

*where $L(\boldsymbol{\theta}) = (\boldsymbol{\theta}^\star - \boldsymbol{\theta})^\top \Sigma (\boldsymbol{\theta}^\star - \boldsymbol{\theta})$.*

*Proof.* We start by expressing the loss function using the linear approximation:

$$\ell(y, \boldsymbol{x}, \boldsymbol{\theta}) = (y - f_{\boldsymbol{\theta}}(\boldsymbol{x}))^2 = \left( \nabla_{\boldsymbol{\theta}} f_{\boldsymbol{\theta}_0}(\boldsymbol{x})^\top (\boldsymbol{\theta}^\star - \boldsymbol{\theta}) \right)^2 .$$

Define $\delta\boldsymbol{\theta} = \boldsymbol{\theta}^\star - \boldsymbol{\theta}$. Then,

$$\ell(y, \boldsymbol{x}, \boldsymbol{\theta}) = \left( \nabla_{\boldsymbol{\theta}} f_{\boldsymbol{\theta}_0}(\boldsymbol{x})^\top \delta\boldsymbol{\theta} \right)^2 .$$

Since $\nabla_{\boldsymbol{\theta}} f_{\boldsymbol{\theta}_0}(\boldsymbol{x}) \sim \mathcal{N}(\boldsymbol{0}, \Sigma)$, it follows that

$$Z = \nabla_{\boldsymbol{\theta}} f_{\boldsymbol{\theta}_0}(\boldsymbol{x})^\top \delta\boldsymbol{\theta} \sim \mathcal{N} \left( 0, \delta\boldsymbol{\theta}^\top \Sigma \delta\boldsymbol{\theta} \right) .$$

Let $\sigma^2 = \delta\boldsymbol{\theta}^\top \Sigma \delta\boldsymbol{\theta}$, so $Z \sim \mathcal{N}(0, \sigma^2)$.

The expected loss $L(\boldsymbol{\theta})$ is

$$L(\boldsymbol{\theta}) = \mathbb{E} \left[ \ell(y, \boldsymbol{x}, \boldsymbol{\theta}) \right] = \mathbb{E} \left[ Z^2 \right] = \sigma^2 .$$

To find the variance of $\ell(y, \boldsymbol{x}, \boldsymbol{\theta})$, we compute

$$\mathrm{Var}_\nu \left[\ell(y, \boldsymbol{x}, \boldsymbol{\theta})\right] = \mathbb{E}_\nu \left[\left(\ell(y, \boldsymbol{x}, \boldsymbol{\theta})\right)^2\right] - \left(\mathbb{E}_\nu \left[\ell(y, \boldsymbol{x}, \boldsymbol{\theta})\right]\right)^2 .$$

Next, compute the fourth moment:

$$\mathbb{E}_\nu \left[\left(\ell(y, \boldsymbol{x}, \boldsymbol{\theta})\right)^2\right] = \mathbb{E}_\nu \left[Z^4\right] .$$

Since $Z$ is normally distributed with mean zero and variance $\sigma^2$, the fourth moment is

$$\mathbb{E}_\nu \left[Z^4\right] = 3\sigma^4 .$$

Therefore, the variance is

$$\begin{aligned}
\mathrm{Var}_\nu \left[\ell(y, \boldsymbol{x}, \boldsymbol{\theta})\right] &= \mathbb{E}_\nu \left[Z^4\right] - \left(\mathbb{E}_\nu \left[Z^2\right]\right)^2 \\
&= 3\sigma^4 - (\sigma^2)^2 \\
&= 3\sigma^4 - \sigma^4 \\
&= 2\sigma^4 \\
&= 2\left(L(\boldsymbol{\theta})\right)^2 .
\end{aligned}$$

And the starndard deviation $\sigma(\boldsymbol{\theta}) = \sqrt{2}L(\boldsymbol{\theta})$

The centered loss is $L(\boldsymbol{\theta}) - \ell(y, \boldsymbol{x}, \boldsymbol{\theta}) = \sigma^2 - Z^2$.

The cumulant generating function $J_{\boldsymbol{\theta}}(\lambda)$ is

$$\begin{aligned}
J_{\boldsymbol{\theta}}(\lambda) &= \ln \mathbb{E} \left[e^{\lambda(L(\boldsymbol{\theta}) - \ell(y, \boldsymbol{x}, \boldsymbol{\theta}))}\right] \\
&= \ln \mathbb{E} \left[e^{\lambda(\sigma^2 - Z^2)}\right] \\
&= \lambda\sigma^2 + \ln \mathbb{E} \left[e^{-\lambda Z^2}\right] .
\end{aligned}$$

Since $Z$ is normally distributed with mean zero and variance $\sigma^2$, the moment generating function of $-Z^2$ is

$$\mathbb{E} \left[e^{-\lambda Z^2}\right] = \frac{1}{\sqrt{1 + 2\lambda\sigma^2}} .$$

Therefore,

$$\begin{aligned}
J_{\boldsymbol{\theta}}(\lambda) &= \lambda\sigma^2 - \frac{1}{2} \ln \left(1 + 2\lambda\sigma^2\right) \\
&= \lambda L(\boldsymbol{\theta}) - \frac{1}{2} \ln \left(1 + 2\lambda L(\boldsymbol{\theta})\right) ,
\end{aligned}$$

which completes the proof. □

**Proposition 6.** Consider a regression problem defined by the mean square error loss, $\ell(y, \boldsymbol{x}, \boldsymbol{\theta}) = (y - f_{\boldsymbol{\theta}}(\boldsymbol{x}))^2$, where $f_{\boldsymbol{\theta}}(\boldsymbol{x})$ represents a regression model implemented by a neural network with parameters $\boldsymbol{\theta}$. The neural network can be *linearized* through a first-order Taylor expansion around the initial parameter configuration $\boldsymbol{\theta}_0$, given by:

$$f_{\boldsymbol{\theta}}(\boldsymbol{x}) \approx f_{\boldsymbol{\theta}_0}(\boldsymbol{x}) + \nabla_{\boldsymbol{\theta}} f_{\boldsymbol{\theta}_0}(\boldsymbol{x})^T (\boldsymbol{\theta} - \boldsymbol{\theta}_0).$$

Assume that, for a given input $\boldsymbol{x}$, the corresponding target value $y$ can be expressed as $y = f_{\boldsymbol{\theta}_0}(\boldsymbol{x}) + \nabla_{\boldsymbol{\theta}} f_{\boldsymbol{\theta}_0}(\boldsymbol{x})^T (\boldsymbol{\theta}^\star - \boldsymbol{\theta}_0)$ for some parameter $\boldsymbol{\theta}^\star \in \boldsymbol{\Theta}$. Additionally, assume that the feature vectors $\nabla_{\boldsymbol{\theta}} f_{\boldsymbol{\theta}_0}(\boldsymbol{x})$ follow a multivariate Normal distribution, $\nabla_{\boldsymbol{\theta}} f_{\boldsymbol{\theta}_0}(\boldsymbol{x}) \sim \mathcal{N}(0, \Sigma)$. Under these assumptions, the gradient of the inverse-rate function can be expressed as:

$$\nabla_{\boldsymbol{\theta}} \mathcal{I}_{\boldsymbol{\theta}}^{-1}(s) = f(s, \boldsymbol{\theta}) \nabla_{\boldsymbol{\theta}} \sigma(\boldsymbol{\theta}),$$

where $f(s, \boldsymbol{\theta})$ is a real-valued function that increases monotonically with $s$.

*Proof.* By Propositon 11, we have that

$$J_{\boldsymbol{\theta}}(\lambda) = \lambda L(\boldsymbol{\theta}) - \frac{1}{2} \ln \left(1 + 2\lambda L(\boldsymbol{\theta})\right) ,$$

By Proposition 9,

$$\nabla_{\boldsymbol{\theta}} \mathcal{I}_{\boldsymbol{\theta}}^{-1}(s) = \frac{1}{\lambda^{\star}} \nabla_{\boldsymbol{\theta}} J_{\boldsymbol{\theta}}(\lambda^{\star}) , \tag{30}$$

where

$$\lambda^{\star} = \arg\min_{\lambda} \frac{J_{\boldsymbol{\theta}}(\lambda) + s}{\lambda} . \tag{31}$$

In this case, we have that

$$\frac{1}{\lambda^{\star}} \nabla_{\boldsymbol{\theta}} J_{\boldsymbol{\theta}}(\lambda^{\star}) = \frac{1}{\lambda^{\star}} \left( \lambda^{\star} \nabla_{\boldsymbol{\theta}} L(\boldsymbol{\theta}) - \frac{\lambda^{\star} \nabla L(\boldsymbol{\theta})}{1 + 2\lambda^{\star} L(\boldsymbol{\theta})} \right) = \nabla L(\boldsymbol{\theta}) \left( 1 - \frac{1}{1 + 2\lambda^{\star} L(\boldsymbol{\theta})} \right)$$

By Proposition 10,

$$\mathrm{Var}_{\nu}\left[\ell(y, \boldsymbol{x}, \boldsymbol{\theta})\right] = 2 \left(L(\boldsymbol{\theta})\right)^{2} ,$$

Rearrging, we have that

$$L(\boldsymbol{\theta}) = \frac{\sigma(\boldsymbol{\theta})}{\sqrt{2}}$$

Finally, combining the above equalities, $\nabla_{\boldsymbol{\theta}} \mathcal{I}_{\boldsymbol{\theta}}^{-1}(s)$ can be written as

$$\nabla_{\boldsymbol{\theta}} \mathcal{I}_{\boldsymbol{\theta}}^{-1}(s) = \sigma(\boldsymbol{\theta}) \frac{1}{\sqrt{2}} \left( 1 - \frac{1}{1 + 2\lambda^{\star} L(\boldsymbol{\theta})} \right)$$

This proof the result defining $f(s, \boldsymbol{\theta})$ as:

$$f(s, \boldsymbol{\theta}) = \frac{1}{\sqrt{2}} \left( 1 - \frac{1}{1 + 2\lambda^{\star} L(\boldsymbol{\theta})} \right)$$

where $\lambda^{\star}$ depends directly on $s$.

$\square$

**Proposition 12.** *For any $\boldsymbol{\theta} \in \Theta$, the inverse rate function $\mathcal{I}_{\boldsymbol{\theta}}^{-1}(s)$ can be expressed as:*

$$\mathcal{I}_{\boldsymbol{\theta}}^{-1}(s) = \nabla_{\lambda} J_{\boldsymbol{\theta}}(\lambda^{*}).$$

*where $\lambda^{*}$ is defined as:*

$$\lambda^{*} = \arg\inf_{\lambda} \left( \frac{s + J_{\boldsymbol{\theta}}(\lambda)}{\lambda} \right).$$

*Proof.* We are given that the inverse rate function $\mathcal{I}_{\boldsymbol{\theta}}^{-1}(s)$ is defined as:

$$\mathcal{I}_{\boldsymbol{\theta}}^{-1}(s) = \inf_{\lambda} \left( \frac{J_{\boldsymbol{\theta}}(\lambda) + s}{\lambda} \right).$$

To find the optimal value $\lambda^{*}$ that minimizes this expression, we differentiate the objective function with respect to $\lambda$ and set the derivative equal to zero:

$$\frac{\partial}{\partial \lambda} \left( \frac{J_{\boldsymbol{\theta}}(\lambda) + s}{\lambda} \right) = 0.$$

First, compute the derivative:

$$\frac{\partial}{\partial \lambda} \left( \frac{J_{\boldsymbol{\theta}}(\lambda) + s}{\lambda} \right) = \frac{\lambda \frac{dJ_{\boldsymbol{\theta}}}{d\lambda} - (J_{\boldsymbol{\theta}}(\lambda) + s)}{\lambda^{2}}.$$

Setting this equal to zero gives the first-order optimality condition:

$$\lambda \frac{dJ_{\boldsymbol{\theta}}}{d\lambda} = J_{\boldsymbol{\theta}}(\lambda) + s.$$

At the optimal point $\lambda^*$, we obtain the relation:

$$\lambda^* \frac{dJ_{\boldsymbol{\theta}}}{d\lambda}\bigg|_{\lambda=\lambda^*} = J_{\boldsymbol{\theta}}(\lambda^*) + s.$$

Thus, solving for $I^{-1}(s)$ in terms of $\lambda^*$ and the gradient of $J_{\boldsymbol{\theta}}(\lambda)$ with respect to $\lambda$, we have:

$$I^{-1}(s) = \nabla_\lambda J_{\boldsymbol{\theta}}(\lambda^*).$$

This concludes the proof. $\qquad\square$

**Proposition 13.** *For any $\boldsymbol{\theta} \in \boldsymbol{\Theta}$, it verifies that*

$$\frac{dJ_{\boldsymbol{\theta}}(\lambda)}{d\lambda} = L(\boldsymbol{\theta}) - \mathbb{E}_{\nu_\lambda}\left[\ell(\boldsymbol{y}, \boldsymbol{x}, \boldsymbol{\theta})\right],$$

*where $\nu_\lambda$ is a tilted probability measure given by*

$$\nu_\lambda(\boldsymbol{y}, \boldsymbol{x}) := \frac{e^{-\lambda \ell(\boldsymbol{y}, \boldsymbol{x}, \boldsymbol{\theta})} \nu(\boldsymbol{y}, \boldsymbol{x})}{\mathbb{E}_\nu\left[e^{-\lambda \ell(\boldsymbol{y}, \boldsymbol{x}, \boldsymbol{\theta})}\right]}.$$

*Proof.* We begin with the definition of the cumulant generating function $J(\lambda)$ as

$$J(\lambda) = \ln Z(\lambda),$$

where $Z(\lambda) = \mathbb{E}_\nu\left[e^{\lambda(L(\boldsymbol{\theta})-\ell(\boldsymbol{y}, \boldsymbol{x}, \boldsymbol{\theta}))}\right]$ is the moment generating function. To compute the gradient of $J(\lambda)$ with respect to $\lambda$, we apply the chain rule:

$$\frac{dJ}{d\lambda} = \frac{1}{Z(\lambda)} \frac{dZ(\lambda)}{d\lambda}.$$

Next, we differentiate $Z(\lambda)$ with respect to $\lambda$:

$$\frac{dZ(\lambda)}{d\lambda} = \mathbb{E}_\nu\left[(L(\boldsymbol{\theta}) - \ell(\boldsymbol{y}, \boldsymbol{x}, \boldsymbol{\theta}))e^{\lambda(L(\boldsymbol{\theta})-\ell(\boldsymbol{y}, \boldsymbol{x}, \boldsymbol{\theta}))}\right].$$

Substituting this result into the expression for $\frac{dJ}{d\lambda}$, we get:

$$\frac{dJ}{d\lambda} = \frac{1}{Z(\lambda)}\mathbb{E}_\nu\left[(L(\boldsymbol{\theta}) - \ell(\boldsymbol{y}, \boldsymbol{x}, \boldsymbol{\theta}))e^{\lambda(L(\boldsymbol{\theta})-\ell(\boldsymbol{y}, \boldsymbol{x}, \boldsymbol{\theta}))}\right].$$

We now introduce the tilted distribution $\nu_\lambda(\boldsymbol{y}, \boldsymbol{x})$, defined as

$$\nu_\lambda(\boldsymbol{y}, \boldsymbol{x}) = \frac{e^{\lambda(L(\boldsymbol{\theta})-\ell(\boldsymbol{y}, \boldsymbol{x}, \boldsymbol{\theta}))}\nu(\boldsymbol{y}, \boldsymbol{x})}{Z(\lambda)} = \frac{e^{-\lambda \ell(\boldsymbol{y}, \boldsymbol{x}, \boldsymbol{\theta})}\nu(\boldsymbol{y}, \boldsymbol{x})}{\mathbb{E}_\nu\left[e^{-\lambda \ell(\boldsymbol{y}, \boldsymbol{x}, \boldsymbol{\theta})}\right]},$$

which allows us to rewrite the expectation as

$$\frac{dJ}{d\lambda} = \mathbb{E}_{\nu_\lambda}\left[L(\boldsymbol{\theta}) - \ell(\boldsymbol{y}, \boldsymbol{x}, \boldsymbol{\theta})\right].$$

Since $L(\boldsymbol{\theta})$ is constant with respect to $\boldsymbol{y}$ and $\boldsymbol{x}$, this simplifies to

$$\frac{dJ}{d\lambda} = L(\boldsymbol{\theta}) - \mathbb{E}_{\nu_\lambda}[\ell(\boldsymbol{y}, \boldsymbol{x}, \boldsymbol{\theta})],$$

which completes the proof. $\qquad\square$

**Proposition 14.** *For any $\boldsymbol{\theta} \in \boldsymbol{\Theta}$, it verifies that*

$$\nabla_{\boldsymbol{\theta}} J_{\boldsymbol{\theta}}(\lambda) = \lambda\left(\nabla_{\boldsymbol{\theta}} L(\boldsymbol{\theta}) - \mathbb{E}_{\nu_\lambda}\left[\nabla_{\boldsymbol{\theta}}\ell(\boldsymbol{y}, \boldsymbol{x}, \boldsymbol{\theta})\right]\right).$$

*where $\nu_\lambda$ is a tilted probability measure given by*

$$\nu_\lambda(\boldsymbol{y}, \boldsymbol{x}) := \frac{e^{-\lambda \ell(\boldsymbol{y}, \boldsymbol{x}, \boldsymbol{\theta})}\nu(\boldsymbol{y}, \boldsymbol{x})}{\mathbb{E}_\nu\left[e^{-\lambda \ell(\boldsymbol{y}, \boldsymbol{x}, \boldsymbol{\theta})}\right]}.$$

*Proof.* To expand the gradient of $J_{\boldsymbol{\theta}}(\lambda)$ with respect to $\boldsymbol{\theta}$, let's start from the definition of $J_{\boldsymbol{\theta}}(\lambda)$. Recall that

$$J_{\boldsymbol{\theta}}(\lambda) = \ln \mathbb{E}_{\nu}\left[e^{\lambda(L(\boldsymbol{\theta})-\ell(\boldsymbol{y},\boldsymbol{x},\boldsymbol{\theta}))}\right],$$

where $L(\boldsymbol{\theta}) = \mathbb{E}_{\nu}[\ell(\boldsymbol{y}, \boldsymbol{x}, \boldsymbol{\theta})]$ is the expected loss, and $(\boldsymbol{y}, \boldsymbol{x}) \sim \nu$. Taking the gradient with respect to $\boldsymbol{\theta}$, we use the chain rule:

$$\nabla_{\boldsymbol{\theta}} J_{\boldsymbol{\theta}}(\lambda) = \frac{1}{\mathbb{E}_{\nu}\left[e^{\lambda(L(\boldsymbol{\theta})-\ell(\boldsymbol{y},\boldsymbol{x},\boldsymbol{\theta}))}\right]} \cdot \nabla_{\boldsymbol{\theta}}\left(\mathbb{E}_{\nu}\left[e^{\lambda(L(\boldsymbol{\theta})-\ell(y,x,\boldsymbol{\theta}))}\right]\right).$$

Now, let's expand $\nabla_{\boldsymbol{\theta}}\left(\mathbb{E}_{\nu}\left[e^{\lambda(L(\boldsymbol{\theta})-\ell(\boldsymbol{y},\boldsymbol{x},\boldsymbol{\theta}))}\right]\right)$:

$$\nabla_{\boldsymbol{\theta}}\mathbb{E}_{\nu}\left[e^{\lambda(L(\boldsymbol{\theta})-\ell(\boldsymbol{y},\boldsymbol{x},\boldsymbol{\theta}))}\right] = \mathbb{E}_{\nu}\left[\nabla_{\boldsymbol{\theta}} e^{\lambda(L(\boldsymbol{\theta})-\ell(\boldsymbol{y},\boldsymbol{x},\boldsymbol{\theta}))}\right].$$

Using the chain rule again on the exponential function, we have:

$$\nabla_{\boldsymbol{\theta}} e^{\lambda(L(\boldsymbol{\theta})-\ell(y,x,\boldsymbol{\theta}))} = e^{\lambda(L(\boldsymbol{\theta})-\ell(y,x,\boldsymbol{\theta}))} \cdot \lambda\left(\nabla_{\boldsymbol{\theta}} L(\boldsymbol{\theta}) - \nabla_{\boldsymbol{\theta}}\ell(y,x,\boldsymbol{\theta})\right).$$

Therefore,

$$\nabla_{\boldsymbol{\theta}} J_{\boldsymbol{\theta}}(\lambda) = \frac{\mathbb{E}_{\nu}\left[e^{\lambda(L(\boldsymbol{\theta})-\ell(y,x,\boldsymbol{\theta}))} \cdot \lambda\left(\nabla_{\boldsymbol{\theta}} L(\boldsymbol{\theta}) - \nabla_{\boldsymbol{\theta}}\ell(\boldsymbol{y},\boldsymbol{x},\boldsymbol{\theta})\right)\right]}{\mathbb{E}_{\nu}\left[e^{\lambda(L(\boldsymbol{\theta})-\ell(\boldsymbol{y},\boldsymbol{x},\boldsymbol{\theta}))}\right]}.$$

We can simplify this expression as

$$\nabla_{\boldsymbol{\theta}} J_{\boldsymbol{\theta}}(\lambda) = \lambda\left(\nabla_{\boldsymbol{\theta}} L(\boldsymbol{\theta}) - \mathbb{E}_{\nu_{\lambda}}\left[\nabla_{\boldsymbol{\theta}}\ell(\boldsymbol{y},\boldsymbol{x},\boldsymbol{\theta})\right]\right),$$

where $\nu_{\lambda}$ is a tilted probability measure given by

$$\nu_{\lambda}(\boldsymbol{y},\boldsymbol{x}) = \frac{e^{\lambda(L(\boldsymbol{\theta})-\ell(\boldsymbol{y},\boldsymbol{x},\boldsymbol{\theta}))}\nu(\boldsymbol{y},\boldsymbol{x})}{\mathbb{E}_{\nu}\left[e^{\lambda(L(\boldsymbol{\theta})-\ell(\boldsymbol{y},\boldsymbol{x},\boldsymbol{\theta}))}\right]} = \frac{e^{-\lambda\ell(\boldsymbol{y},\boldsymbol{x},\boldsymbol{\theta})}\nu(\boldsymbol{y},\boldsymbol{x})}{\mathbb{E}_{\nu}\left[e^{-\lambda\ell(\boldsymbol{y},\boldsymbol{x},\boldsymbol{\theta})}\right]}.$$

Thus, the gradient of $J_{\boldsymbol{\theta}}(\lambda)$ with respect to $\boldsymbol{\theta}$ is

$$\nabla_{\boldsymbol{\theta}} J_{\boldsymbol{\theta}}(\lambda) = \lambda\left(\nabla_{\boldsymbol{\theta}} L(\boldsymbol{\theta}) - \mathbb{E}_{\nu_{\lambda}}\left[\nabla_{\boldsymbol{\theta}}\ell(\boldsymbol{y},\boldsymbol{x},\boldsymbol{\theta})\right]\right).$$

□

**Proposition 15.** *For any $\boldsymbol{\theta} \in \Theta$, it verifies that*

$$\nabla_{\boldsymbol{\theta}}\mathcal{I}_{\boldsymbol{\theta}}^{-1}(s) = \nabla_{\boldsymbol{\theta}} L(\boldsymbol{\theta}) - \mathbb{E}_{\nu_{\lambda^{\star}}}\left[\nabla_{\boldsymbol{\theta}}\ell(\boldsymbol{y},\boldsymbol{x},\boldsymbol{\theta})\right]$$

*where $\lambda^{\star} := \arg\min_{\lambda} \frac{s+J_{\boldsymbol{\theta}}(\lambda)}{\lambda}$.*

*Proof.* Given that the inverse rate can be written as

$$\mathcal{I}_{\boldsymbol{\theta}}^{-1}(s) = G_{\boldsymbol{\theta}}(s, \lambda^{\star}(\boldsymbol{\theta})),$$

where

$$G_{\boldsymbol{\theta}}(s, \lambda) := \frac{s + J_{\boldsymbol{\theta}}(\lambda)}{\lambda},$$

and

$$\lambda^{\star}(\boldsymbol{\theta}) = \arg\min_{\lambda} \frac{s + J_{\boldsymbol{\theta}}(\lambda)}{\lambda},$$

we want to compute the gradient $\nabla_{\boldsymbol{\theta}}\mathcal{I}_{\boldsymbol{\theta}}^{-1}(s)$. Using the chain rule:

$$\nabla_{\boldsymbol{\theta}}\mathcal{I}_{\boldsymbol{\theta}}^{-1}(s) = \left.\frac{\partial G_{\boldsymbol{\theta}}(s, \lambda)}{\partial \lambda}\right|_{\lambda=\lambda^{\star}(\boldsymbol{\theta})} \cdot \frac{\partial \lambda^{\star}(\boldsymbol{\theta})}{\partial \boldsymbol{\theta}} + \left.\frac{\partial G_{\boldsymbol{\theta}}(s, \lambda)}{\partial \boldsymbol{\theta}}\right|_{\lambda=\lambda^{\star}(\boldsymbol{\theta})}.$$

Since $\lambda^{\star}(\boldsymbol{\theta})$ minimizes $G_{\boldsymbol{\theta}}(s, \lambda)$, the derivative with respect to $\lambda$ is zero at $\lambda = \lambda^{\star}(\boldsymbol{\theta})$. The expression simplifies to:

$$\nabla_{\boldsymbol{\theta}}\mathcal{I}_{\boldsymbol{\theta}}^{-1}(s) = \left.\frac{\partial G_{\boldsymbol{\theta}}(s, \lambda)}{\partial \boldsymbol{\theta}}\right|_{\lambda=\lambda^{\star}(\boldsymbol{\theta})}.$$

Since $G_{\boldsymbol{\theta}}(s, \lambda) = \frac{s+J_{\boldsymbol{\theta}}(\lambda)}{\lambda}$, differentiating with respect to $\boldsymbol{\theta}$ yields:

$$\nabla_{\boldsymbol{\theta}}\mathcal{I}_{\boldsymbol{\theta}}^{-1}(s) = \frac{\nabla_{\boldsymbol{\theta}} J_{\boldsymbol{\theta}}(\lambda^{\star}(\boldsymbol{\theta}))}{\lambda^{\star}(\boldsymbol{\theta})}.$$

Using Proposition 14 we have that

$$\nabla_{\boldsymbol{\theta}}\mathcal{I}_{\boldsymbol{\theta}}^{-1}(s) = \nabla_{\boldsymbol{\theta}} L(\boldsymbol{\theta}) - \mathbb{E}_{\nu_{\lambda^{\star}}}\left[\nabla_{\boldsymbol{\theta}}\ell(\boldsymbol{y},\boldsymbol{x},\boldsymbol{\theta})\right]$$

□

**Proposition 16.** *Let $\nu_\lambda(\boldsymbol{y}, \boldsymbol{x})$ be the tilted distribution defined as*

$$\nu_\lambda(\boldsymbol{y}, \boldsymbol{x}) = \frac{e^{-\lambda \ell(\boldsymbol{y}, \boldsymbol{x}, \boldsymbol{\theta})} \nu(\boldsymbol{y}, \boldsymbol{x})}{\mathbb{E}_\nu \left[ e^{-\lambda \ell(\boldsymbol{y}, \boldsymbol{x}, \boldsymbol{\theta})} \right]} \,.$$

*Then, the cumulant generating function $J_{\boldsymbol{\theta}}(\lambda)$ can be expressed as the Kullback-Leibler divergence between $\nu$ and $\nu_\lambda$:*

$$J_{\boldsymbol{\theta}}(\lambda) = \mathrm{KL}(\nu \,\|\, \nu_\lambda) \,.$$

*Proof.* We start by computing the KL divergence $\mathrm{KL}(\nu \,\|\, \nu_\lambda)$:

$$\mathrm{KL}(\nu \,\|\, \nu_\lambda) = \int \nu(\boldsymbol{y}, \boldsymbol{x}) \ln \left( \frac{\nu(\boldsymbol{y}, \boldsymbol{x})}{\nu_\lambda(\boldsymbol{y}, \boldsymbol{x})} \right) d\boldsymbol{y} \, d\boldsymbol{x}$$

$$= \int \nu(\boldsymbol{y}, \boldsymbol{x}) \ln \left( \frac{\nu(\boldsymbol{y}, \boldsymbol{x})}{\frac{e^{-\lambda \ell(\boldsymbol{y}, \boldsymbol{x}, \boldsymbol{\theta})} \nu(\boldsymbol{y}, \boldsymbol{x})}{\mathbb{E}_\nu \left[ e^{-\lambda \ell(\boldsymbol{y}, \boldsymbol{x}, \boldsymbol{\theta})} \right]}} \right) d\boldsymbol{y} \, d\boldsymbol{x}$$

$$= \int \nu(\boldsymbol{y}, \boldsymbol{x}) \ln \left( \frac{\mathbb{E}_\nu \left[ e^{-\lambda \ell(\boldsymbol{y}, \boldsymbol{x}, \boldsymbol{\theta})} \right]}{e^{-\lambda \ell(\boldsymbol{y}, \boldsymbol{x}, \boldsymbol{\theta})}} \right) d\boldsymbol{y} \, d\boldsymbol{x}$$

$$= \ln \mathbb{E}_\nu \left[ e^{-\lambda \ell(\boldsymbol{y}, \boldsymbol{x}, \boldsymbol{\theta})} \right] + \lambda \int \nu(\boldsymbol{y}, \boldsymbol{x}) \ell(\boldsymbol{y}, \boldsymbol{x}, \boldsymbol{\theta}) \, d\boldsymbol{y} \, d\boldsymbol{x}$$

$$= \ln \mathbb{E}_\nu \left[ e^{-\lambda \ell(\boldsymbol{y}, \boldsymbol{x}, \boldsymbol{\theta})} \right] + \lambda L(\boldsymbol{\theta}) \,.$$

Recall that the cumulant generating function $J(\lambda)$ can be rewritten as:

$$J(\lambda) = \ln \mathbb{E}_\nu \left[ e^{\lambda (L(\boldsymbol{\theta}) - \ell(\boldsymbol{y}, \boldsymbol{x}, \boldsymbol{\theta}))} \right]$$

$$= \ln \left( e^{\lambda L(\boldsymbol{\theta})} \mathbb{E}_\nu \left[ e^{-\lambda \ell(\boldsymbol{y}, \boldsymbol{x}, \boldsymbol{\theta})} \right] \right)$$

$$= \lambda L(\boldsymbol{\theta}) + \ln \mathbb{E}_\nu \left[ e^{-\lambda \ell(\boldsymbol{y}, \boldsymbol{x}, \boldsymbol{\theta})} \right] \,.$$

Comparing the expressions for $\mathrm{KL}(\nu \,\|\, \nu_\lambda)$ and $J(\lambda)$, we find that:

$$\mathrm{KL}(\nu \,\|\, \nu_\lambda) = \lambda L(\boldsymbol{\theta}) + \ln \mathbb{E}_\nu \left[ e^{-\lambda \ell(\boldsymbol{y}, \boldsymbol{x}, \boldsymbol{\theta})} \right] = J(\lambda) \,.$$

Therefore,

$$J(\lambda) = \mathrm{KL}(\nu \,\|\, \nu_\lambda) \,,$$

which completes the proof. $\qquad \square$

**Proposition 17.** *For any $a' \geq a \geq 0$, it holds that*

$$J_{\boldsymbol{\theta}}(\lambda^\star(a')) \geq J_{\boldsymbol{\theta}}(\lambda^\star(a)) \,.$$

*where*

$$\lambda^\star(\alpha) = \arg\sup_\lambda \lambda \alpha - J_{\boldsymbol{\theta}}(\lambda) \,,$$

*Proof.* Since $J_{\boldsymbol{\theta}}(\lambda)$ is convex and differentiable, its derivative $\nabla_\lambda J_{\boldsymbol{\theta}}(\lambda)$ exists and is monotonically increasing. The Legendre transform relates $\lambda^\star(a)$ and $a$ via the derivative of $J$:

$$\nabla_\lambda J_{\boldsymbol{\theta}}(\lambda^\star(a)) = a \,.$$

Similarly, for $a'$,

$$\nabla_\lambda J_{\boldsymbol{\theta}}(\lambda^\star(a')) = a' \,.$$

Given that $a' \geq a$ and $\nabla_\lambda J(\lambda)$ is increasing, it follows that

$$\nabla_\lambda J_{\boldsymbol{\theta}}(\lambda^\star(a')) = a' \geq a = \nabla_\lambda J_{\boldsymbol{\theta}}(\lambda^\star(a)) \,.$$

Therefore,

$$\lambda^\star(a') \geq \lambda^\star(a) \,.$$

Now, since $J(\lambda)$ is convex, it satisfies the property that for any $\lambda_1 \leq \lambda_2$,

$$J_{\boldsymbol{\theta}}(\lambda_1) \leq J_{\boldsymbol{\theta}}(\lambda_2).$$

Applying this property to $\lambda^\star(a)$ and $\lambda^\star(a')$, we have

$$J_{\boldsymbol{\theta}}(\lambda^\star(a')) \geq J_{\boldsymbol{\theta}}(\lambda^\star(a)).$$

This completes the proof. $\qquad\square$

**Proposition 18.** *Let $\nu_\lambda(\boldsymbol{y}, \boldsymbol{x})$ be the tilted distribution defined as*

$$\nu_\lambda(\boldsymbol{y}, \boldsymbol{x}) = \frac{e^{-\lambda\ell(\boldsymbol{y}, \boldsymbol{x}, \boldsymbol{\theta})}\nu(\boldsymbol{y}, \boldsymbol{x})}{\mathbb{E}_\nu\left[e^{-\lambda\ell(\boldsymbol{y}, \boldsymbol{x}, \boldsymbol{\theta})}\right]}.$$

*if the loss function $\ell(\boldsymbol{y}, \boldsymbol{x}, \boldsymbol{\theta})$ is $M$-Lipschitz with respect to $(\boldsymbol{y}, \boldsymbol{x})$, then,*

$$\|\nabla_{\boldsymbol{\theta}} L(\boldsymbol{\theta}) - \mathbb{E}_{\nu_\lambda}[\nabla_{\boldsymbol{\theta}}\ell(\boldsymbol{y}, \boldsymbol{x}, \boldsymbol{\theta})]\| \leq M\sqrt{2D_{KL}(\nu\|\nu\lambda)}.$$

*Proof.* Let us rewrite the difference in expectations:

$$\|E_\nu[\nabla_{\boldsymbol{\theta}}\ell(y, x, \boldsymbol{\theta})] - E_{\nu\lambda^\star}[\nabla_{\boldsymbol{\theta}}\ell(y, x, \boldsymbol{\theta})]\| = \left\|\int \nabla_{\boldsymbol{\theta}}\ell(y, x, \boldsymbol{\theta})\left(\nu(y, x) - \nu\lambda(y, x)\right) dy\, dx\right\|$$

By applying Hölder's inequality, we can bound this by:

$$\left\|\int \nabla_{\boldsymbol{\theta}}\ell(y, x, \boldsymbol{\theta})\left(\nu(y, x) - \nu\lambda(y, x)\right) dy\, dx\right\| \leq \int \|\nabla_{\boldsymbol{\theta}}\ell(y, x, \boldsymbol{\theta})\|\,|\nu(y, x) - \nu\lambda(y, x)|\;dy\, dx$$

Notice that the total variation distance between $\nu$ and $\nu\lambda$, defined as

$$d_{TV}(\nu, \nu\lambda) = \frac{1}{2}\int |\nu(y, x) - \nu\lambda(y, x)|\, dy\, dx$$

The bound then becomes:

$$\|E_\nu[\nabla_{\boldsymbol{\theta}}\ell(y, x, \boldsymbol{\theta})] - E_{\nu\lambda}[\nabla_{\boldsymbol{\theta}}\ell(y, x, \boldsymbol{\theta})]\| \leq \sup_{(y,x)} \|\nabla_{\boldsymbol{\theta}}\ell(y, x, \boldsymbol{\theta})\| \cdot 2d_{TV}(\nu, \nu\lambda)$$

Pinsker's inequality states that for two probability densities $\nu$ and $\nu\lambda$,

$$d_{TV}(\nu, \nu\lambda) \leq \sqrt{\frac{1}{2}D_{KL}(\nu\|\nu\lambda^\star)}.$$

Chaining Pinsker's inequality into the above bound gives:

$$\|E_\nu[\nabla_{\boldsymbol{\theta}}\ell(y, x, \boldsymbol{\theta})] - E_{\nu\lambda}[\nabla_{\boldsymbol{\theta}}\ell(y, x, \boldsymbol{\theta})]\| \leq \sup_{(y,x)} \|\nabla_{\boldsymbol{\theta}}\ell(y, x, \boldsymbol{\theta})\| \cdot 2\sqrt{\frac{1}{2}D_{KL}(\nu\|\nu\lambda)}.$$

The bound can be further simplified as:

$$\|E_\nu[\nabla_{\boldsymbol{\theta}}\ell(y, x, \boldsymbol{\theta})] - E_{\nu\lambda}[\nabla_{\boldsymbol{\theta}}\ell(y, x, \boldsymbol{\theta})]\| \leq \sup_{(y,x)} \|\nabla_{\boldsymbol{\theta}}\ell(y, x, \boldsymbol{\theta})\| \cdot \sqrt{2D_{KL}(\nu\|\nu\lambda)}.$$

Finally, assuming that the loss function $\ell(y, x, \boldsymbol{\theta})$ is M-Lipchitz, we prove the inequality, because in this case we have

$$\sup_{(y,x)} \|\nabla_{\boldsymbol{\theta}}\ell(y, x, \boldsymbol{\theta})\| \leq M$$

$\qquad\square$

**Proposition 19.** *For any $\boldsymbol{\theta} \in \Theta$, $n > 0$ and $D \sim \nu^n$, there exists a distribution $\gamma(\boldsymbol{y}, \boldsymbol{x})$ that depends on $\boldsymbol{\theta}$ and $\alpha(D, \boldsymbol{\theta})$, such that,*

$$\nabla_{\boldsymbol{\theta}}\hat{L}(D, \boldsymbol{\theta}) = \nabla_{\boldsymbol{\theta}}\mathbb{E}_\gamma[\ell(\boldsymbol{y}, \boldsymbol{x}, \boldsymbol{\theta})] \tag{32}$$

*Proof.*  □

**Theorem 7** For any $\boldsymbol{\theta} \in \boldsymbol{\Theta}$, $n > 0$ and $D \sim \nu^n$, there exists a distribution $\gamma(\boldsymbol{y}, \boldsymbol{x})$ that depends on $\boldsymbol{\theta}$ and $\alpha(D, \boldsymbol{\theta})$, such that,

$$\nabla_{\boldsymbol{\theta}} \hat{L}(D, \boldsymbol{\theta}) = \nabla_{\boldsymbol{\theta}} \mathbb{E}_{\gamma}[\ell(\boldsymbol{y}, \boldsymbol{x}, \boldsymbol{\theta})] \tag{33}$$

and KL $(\nu \mid \gamma)$ is monotonically increasing with $\alpha(D, \boldsymbol{\theta})$ and KL $(\nu \mid \gamma) = 0$ if $\alpha(D, \boldsymbol{\theta}) = 0$. Furthermore, if the loss function $\ell(\boldsymbol{y}, \boldsymbol{x}, \boldsymbol{\theta})$ is $M$-Lipschitz with respect to $(\boldsymbol{y}, \boldsymbol{x})$. Then,

$$\|\nabla_{\boldsymbol{\theta}} \mathcal{I}_{\boldsymbol{\theta}}^{-1}(s)_{|s=\alpha(D,\boldsymbol{\theta})} \|_2 \le M \sqrt{2 \, \text{KL}(\nu \mid \gamma)}. \tag{34}$$

*Proof.* **Part I:** Let us start proving that

$$\nabla_{\boldsymbol{\theta}} \hat{L}(D, \boldsymbol{\theta}) = \nabla_{\boldsymbol{\theta}} \mathbb{E}_{\gamma}[\ell(\boldsymbol{y}, \boldsymbol{x}, \boldsymbol{\theta})]$$

From Proposition 12, we have that

$$\mathcal{I}_{\boldsymbol{\theta}}^{-1}(\alpha(D, \boldsymbol{\theta})) = \nabla_{\lambda} J_{\boldsymbol{\theta}}(\lambda^*).$$

where $\lambda^*$ is defined as:

$$\lambda^* = \arg\inf_{\lambda} \left( \frac{\alpha(D, \boldsymbol{\theta}) + J_{\boldsymbol{\theta}}(\lambda)}{\lambda} \right).$$

From Proposition 13, we have that

$$\mathcal{I}_{\boldsymbol{\theta}}^{-1}(\alpha(D, \boldsymbol{\theta})) = \nabla_{\lambda} J_{\boldsymbol{\theta}}(\lambda^{\star}) = L(\boldsymbol{\theta}) - \mathbb{E}_{\nu\lambda^*}[\ell(\boldsymbol{y}, \boldsymbol{x}, \boldsymbol{\theta})],$$

where $\nu\lambda^*$ is a tilted probability measure given by

$$\nu\lambda^*(\boldsymbol{y}, \boldsymbol{x}) := \frac{e^{-\lambda^* \ell(\boldsymbol{y}, \boldsymbol{x}, \boldsymbol{\theta})} \nu(\boldsymbol{y}, \boldsymbol{x})}{\mathbb{E}_{\nu}\left[e^{-\lambda^* \ell(\boldsymbol{y}, \boldsymbol{x}, \boldsymbol{\theta})}\right]}.$$

From Proposition 3 we have that

$$\hat{L}(D, \boldsymbol{\theta}) = L(\boldsymbol{\theta}) - \mathcal{I}_{\boldsymbol{\theta}}^{-1}(\alpha(D, \boldsymbol{\theta})).$$

Replacing the above terms,

$$\hat{L}(D, \boldsymbol{\theta}) = L(\boldsymbol{\theta}) - \left(L(\boldsymbol{\theta}) - \mathbb{E}_{\nu\lambda^*}[\ell(\boldsymbol{y}, \boldsymbol{x}, \boldsymbol{\theta})]\right).$$

Simplifying, we arrive to

$$\hat{L}(D, \boldsymbol{\theta}) = \mathbb{E}_{\nu\lambda^*}[\ell(\boldsymbol{y}, \boldsymbol{x}, \boldsymbol{\theta})]$$

where the titled distribution $\nu\lambda^{\star}$ is the distribution $\gamma$ referred in the statement of the theorem.

**Part II:** Here we will prove that KL $(\nu \mid \gamma)$ is monotonically increasing with $\alpha(D, \boldsymbol{\theta})$ and KL $(\nu \mid \gamma) = 0$ if $\alpha(D, \boldsymbol{\theta}) = 0$.

By Proposition 16, we have that

$$\text{KL}(\nu \| \nu\lambda^{\star}) = J_{\boldsymbol{\theta}}(\lambda^{\star}).$$

And Proposition 17 states that any $a' \ge a$, it holds that

$$J_{\boldsymbol{\theta}}(\lambda^{\star}(a')) \ge J_{\boldsymbol{\theta}}(\lambda^{\star}(a)).$$

From here we deduce that if the level of abnormality for one data set $D'$ is higher than for other dataset $D$, i..e, $\alpha(D', \boldsymbol{\theta}) > \alpha(D, \boldsymbol{\theta})$, then

$$J_{\boldsymbol{\theta}}(\lambda^{\star}(\alpha(D', \boldsymbol{\theta}))) \ge J_{\boldsymbol{\theta}}(\lambda^{\star}(\alpha(D, \boldsymbol{\theta}))).$$

From where we can deduce that the KL$(\nu \| \nu\lambda^{\star})$ is monotonically increasing with the level of abnormality.

Finally, we have that if $\alpha(D, \boldsymbol{\theta}) = 0$, then KL$(\nu \| \nu\lambda^{\star}) = 0$

Since $J_{\boldsymbol{\theta}}(\lambda)$ is convex and differentiable, its derivative $\nabla_{\lambda} J_{\boldsymbol{\theta}}(\lambda)$ exists and is monotonically increasing. The Legendre transform relates $\lambda^{\star}(a)$ and $a$ via the derivative of $J$:

$$\nabla_{\lambda} J_{\boldsymbol{\theta}}(\lambda^{\star}(a)) = a\,.$$

Then, we deduce that

$$\lambda^{\star}(0) = 0$$

In consequence, when $\alpha(D, \boldsymbol{\theta}) = 0$, $\nu \lambda^{\star}$ will be equal to $\nu$ and the KL divergence between $\nu$ and $\nu \lambda^{\star}$ is equal to zero.

**Part III:** Here we prove that if the loss function $\ell(\boldsymbol{y}, \boldsymbol{x}, \boldsymbol{\theta})$ is $M$-Lipschitz with respect to $(\boldsymbol{y}, \boldsymbol{x})$. Then,

$$\|\nabla_{\boldsymbol{\theta}} \mathcal{I}_{\boldsymbol{\theta}}^{-1}(s)_{|s=\alpha(D,\boldsymbol{\theta})}\|_2 \le M \sqrt{2 \,\text{KL}\,(\nu \mid \gamma)}\,.$$

By Proposition 15, we have

$$\nabla_{\boldsymbol{\theta}} \mathcal{I}_{\boldsymbol{\theta}}^{-1}(s)_{|s=\alpha(D,\boldsymbol{\theta})} = \nabla_{\boldsymbol{\theta}} L(\boldsymbol{\theta}) - \mathbb{E}_{\nu_{\lambda^{\star}}} \left[\nabla_{\boldsymbol{\theta}} \ell(\boldsymbol{y}, \boldsymbol{x}, \boldsymbol{\theta})\right]$$

By Proposition 18, and using that the loss function $\ell(\boldsymbol{y}, \boldsymbol{x}, \boldsymbol{\theta})$ is $M$-Lipschitz with respect to $(\boldsymbol{y}, \boldsymbol{x})$, then,

$$\|\nabla_{\boldsymbol{\theta}} L(\boldsymbol{\theta}) - \mathbb{E}_{\nu \lambda^{*}}[\nabla_{\boldsymbol{\theta}} \ell(\boldsymbol{y}, \boldsymbol{x}, \boldsymbol{\theta})]\| \le M \sqrt{2 D_{KL}(\nu \| \nu \lambda^{*})}\,.$$

By combining the last two inequalites, we finalize the proof.

$\square$

