# OpenReview forum: "A Large Deviation Theory Analysis on the Implicit Bias of SGD"
_ICLR.cc/2025/Conference — ICLR 2025 Conference Withdrawn Submission_

### Official Review · Reviewer_f4No · 2024-10-29

**Soundness:** 3
**Presentation:** 3
**Contribution:** 3
**Rating:** 6
**Confidence:** 4

**Summary:**

This paper applies the large deviation theory to study the generalization error of mini-batch SGD, in turn, provides a perspective on the implicit regularization and the generalization ability.

**Strengths:**

The paper is well-organized and well-written. The understanding of generalization ability from the perspective of LDT is interesting and the findings are novel. It also hints at a possible variant of SGD by introducing the "skip".

**Weaknesses:**

See **Questions**.

**Questions:**

In the following, there are several questions to be addressed.

**Major**
1. About the large index $n$. What value of $n$ can be regarded as a large $n$ in practice? In Line 210, $n$ takes a value of 50,000 and the authors state that this is "a universal cut-off for any model and any data-generating distribution". What about a dataset with less samples, say 30,000 or even less? Will the LDT analysis still apply?

2. On the other hand, to plot Figure 1 (right), $n$ takes a rather small value of 50. Equation (9) seems to be a good prediction. However, 50 is not *large*. Therefore again my question, when can we call $n$ a large number? Is there any criterion, at least qualitatively?

3. The above questions are also intimately related to Equation (12) about the decomposition of the loss gradient with *mini-batches* which are typically not large in machine learning as stated by the authors in Line 309. For example, online learning uses a batch size of 1; many small-scale experiments use batch sizes of 32 or 64, which are not large compared with the total number of data in the dataset. If we cannot be confident about whether a value of $n$ is large or not, it will be difficult to apply the theories developed here. Please elaborate more on this.

4. From Line 244 to Line 298, this section is entitled with *GD*, without mini-batch sampling. However, in the discussion of Figure 2, different batch sizes are taken. I feel confused about this section. If this section is indeed meant to be about GD, could you please explain why batch sizes are mentioned and how they relate to GD?

**Minor**
1. In Figure 1 (left), three instances of the empirical loss Inception V3 model are plotted. What is the horizontal axis? I suppose that $\mathbf{\theta}$ is a vector rather than a scalar. What are those vertical dashed lines? Another question is that from the figure, it seems that the three instances have different expectation values of the loss, in contract to the statement in Line 064 that "each model's empirical loss $\hat{L}_n(\mathbf{\theta})$ has mean equal to $L(\mathbf{\theta})$. While I understand the statement, how may I understand this figure?

2. In Theorem 1, why does the rate function appear with an absolute value in the r.h.s while it has already been signed to be positive? Also, in Figure 1 (center), the rate function may take negative values. Why?

---

### Official Review · Reviewer_FLQi · 2024-11-02

**Soundness:** 2
**Presentation:** 2
**Contribution:** 1
**Rating:** 3
**Confidence:** 3

**Summary:**

This paper introduces tools from "large deviation theory" to try to explain the beneficial implicit bias of SGD.  They decompose the generalization gap of a machine learning model into: (a) the "abnormality" of the training dataset (which measures how unrepresentative the loss on the training dataset is, relative to the loss on randomly sampled datasets of the same size), and (b) the degree to which the loss on randomly sampled datasets of this size is concentrated. They argue that this decomposition is informative.

**Strengths:**

The paper introduces a three-way decomposition of both the full-batch gradient and the stochastic gradient, and shows empirically that a certain component in this decomposition is highly similar across the two gradients.   This is a non-obvious finding.  (However, I have questions about this result - see the 'questions' section.)

More generally, the paper offers a very original take on an important question (the implicit bias of SGD).

**Weaknesses:**

- Theorems 1,2,4 and proposition 5 consider fixed parameters $\theta$ and study the behavior of the empirical loss under random draws of a dataset $D$. However, in machine learning, the training dataset $D$ is not sampled independent of $\theta$.  This is indeed the core challenge of the whole field of learning theory.  Thus, I don't understand why Theorems 1,2,4 and proposition 5 are relevant.
- The paper does not actually attempt to provide an full explanation for why SGD leads to better generalization, relative to GD.  Instead, the paper merely attempts to shed light on the _mechanism_ by which SGD leads to improved generalization.  While this is certainly a valuable goal, I cannot actually follow the paper's logic (see below).
- The crux of the paper is to advocate for decomposition (3) as a meaningful decomposition of the generalization gap.  This decomposition decomposes the generalization gap into: (a) the "abnormality" of the training dataset (which measures how unrepresentative the loss on the training dataset is, relative to the loss on randomly sampled datasets of the same size), and (b) the degree to which the loss on randomly sampled datasets of this size is concentrated.  Lines 323 - 365 seem to be arguing that the SGD's efficacy can be empirically localized to the _first_ of these factors (the "abnormality"), as opposed to the second (the "concentration").  However, the experiments in Figure 2 show that relative to GD, SGD improves _both_ abnormality and concentration.  This seems to contradict the argument in lines 323 - 365 that SGD's success is due to the abnormality alone.  This suggests that the decomposition (3) is not a particularly enlightening way of reasoning about the generalization gap.
  - Line 469 seems to be offering an explanation for why SGD also improves the concentration of the loss, but I find the argument convoluted and cannot follow it.  To me, the occam's razor explanation for why SGD improves both concentration and abnormality is that this was not an informative decomposition of the generalization gap in the first place.
 - Lines 471-486 show that if we have access to the test set, we can use this information to skip certain SGD steps and therefore generalize better.  This seems obvious to me, and does not strike me as supporting the paper's theoretical analysis, as claimed on line 482. Further, I would point out that Figure 5 shows that this intervention also improves 'concentration', not just 'abnormality' (which it is intended to do), which further calls into question the utility of this decomposition.

Overall, I'm not convinced that the paper sheds _any_ light on why SGD generalizes better than GD.

**Questions:**

In the decomposition of the gradient and stochastic gradient in equations 11 and 12, how large are each of the three terms?  The paper emphasizes that the second term tends to be the highly aligned between GD and SGD, which is indeed interesting, but I am wondering whether this term might be smaller in norm than the other two.

---

### Official Review · Reviewer_r3m1 · 2024-11-04

**Soundness:** 2
**Presentation:** 3
**Contribution:** 2
**Rating:** 3
**Confidence:** 2

**Summary:**

this paper uses LDT framework to analyze the generalization error and to characterize the implicit bias of SGD. First the paper provides an LDT-centric view on the generalization gap, treating it as a random variable over dataset draw.  In particular, the paper provides a decomposition of empirical loss along the lines of LDT where they split it into the expected loss and a function of a generalization gap. The paper then proceeds to characterize the gradient of the empirical loss according to this decomposition and provides an explanation for the regularizing effect of mini-batch SGD.

**Strengths:**

I find the framework of decomposing the empirical loss and its gradient into distinct components compelling, with each component addressing specific contributions—one aligned with minimizing the expected loss, while the remaining terms account for deviations. The paper’s additional partitioning of these deviations into two parts is also notable: one that remains consistent between gradient descent (GD) and stochastic gradient descent (SGD), thereby isolating the component responsible for explaining the generalization gap, identified as the gradient of abnormality in generalization error.

**Weaknesses:**

Firstly, I have some reservations regarding the suitability of the chosen formulation of Large Deviation Theory (LDT) for addressing the problem of generalization error. Specifically, in the concentration inequalities ((2) and onwards), the probability is considered over the dataset draw while the parameters are held fixed. Informally, it seems likely that the training loss on a given dataset, at the end of training, will typically fall at the far left of the distribution of empirical losses across dataset draws. In some cases, it may be an outlier—a ‘one-off point’ on the left—while the remainder of the distribution might shift significantly to the right and center around the expected loss.
Given that LDT is inherently a probabilistic framework aimed at describing the behavior of the tails of distributions, it may not be entirely suited for capturing an outlier that lies apart from the distribution. However, it is precisely the difference between the empirical loss on the training dataset and the expected loss that is of primary interest here.
To illustrate, consider an overparameterized neural network setting where training continues until convergence, achieving zero training loss. In these cases, the training data often becomes overfitted—though it can still be beneficial to proceed with training even after the network ‘begins to overfit’ (i.e., when training and validation loss diverge). Once the parameters are fixed, we can evaluate the empirical loss on alternative dataset draws. Assuming the data is sufficiently high-dimensional or exhibits some level of sparsity (such that memorizing the training data alone does not generalize well), evaluating on other dataset draws could approximate the validation or test loss.
In scenarios where limited regularization is applied and the model is forced to overfit (for instance, by gradient flow), performance on these 'validation' sets may approach randomness. This would result in a distribution of empirical losses that centers around the expected loss, with the empirical loss on the specific training dataset at zero—thus representing the aforementioned outlier or 'one-off point' within the distribution. However, the actual distributions and the position of the training loss on a given dataset may differ significantly from this assumption.
It appears that the issue may stem from the fact that LDT provides bounds for $\hat L(D, \theta)$ rather than $min_\theta \hat L(D, \theta)$. While I could be mistaken, I remain uncertain that this distinction is inconsequential. Given the circumstances described, I am not fully sure that LDT offers meaningful insights into the generalization error.
On a somewhat contrasting note, the paper includes experimental examples using datasets with only 50 samples. While I understand this choice may be due to computational constraints, I’m uncertain that it accurately reflects typical deep learning scenarios where models learn specific data representations. Consequently, I’m hesitant to conclude that the distribution graphs presented are truly representative of what occurs in broader deep learning contexts.
It might be the case that LDT may serve as a heuristic for decomposing empirical loss in a meaningful way. As mentioned earlier, I think this is a strength of the paper. However, in the specific context of this paper, I am not fully sure that this reformulation offers meaningful insights into the nature of generalization error. For example, simplifying equation (4), the inverse of the rate function approximates a square root (as shown in equation (5)), suggesting that the 'abnormality' of generalization error is approximated by the square of the generalization loss. In this light, it is unclear to me why high abnormality isn’t simply another way of indicating high generalization loss, as stated in the paper; the added focus on abnormality doesn’t immediately seem to provide further benefits beyond analyzing generalization error directly.
This concern extends to the observation in the section “SGD PREVENTS HIGHLY ABNORMAL GENERALIZATION ERRORS” (line 402), where it is suggested that smaller batch sizes lead to reduced cosine similarity between the gradient of abnormality for the full dataset and for the batch. If I understand correctly, this point is illustrated solely in Figure 3. With this in mind, it feels though this is a sophisticated reformulation of the generalization error, but lacks sufficient justification for the claim that mini-batch gradient computations lead to smaller generalization errors.
Furthermore, if LDT’s role here is largely heuristic, I believe its value would be more convincing with a broader set of experiments.

**Questions:**

- Would you mind clarifying how the LDT deals with the case of the training loss being an outlier in the distribution of empirical losses, or why this is not the case in this data?
- Would you mind providing more explanation about the regularizing effect of SGD in the light of your framework? In particular, I am curious about the effect of cosine similarity of the gradient of the abnormality as we vary the batch size

---

### Note · Authors · 2024-11-13

I have read and agree with the venue's withdrawal policy on behalf of myself and my co-authors.